# World-aware Planning Narratives Enhance Large Vision-Language Model Planner

**Junhao Shi[1,2]\*, Zhaoye Fei[1]\*, Siyin Wang[1,2], Qipeng Guo[2,3],**
**Jingjing Gong[2]†, Xipeng Qiu[1,2]†**

[1]Fudan University  [2]Shanghai Innovation Institute  [3]Shanghai AI Laboratory
jhshi24@m.fudan.edu.cn, zyfei20@fudan.edu.cn
jjgongjj@gmail.com, xpqiu@fudan.edu.cn

## Abstract

Large Vision-Language Models (LVLMs) show promise for embodied planning tasks but struggle with complex scenarios involving unfamiliar environments and multi-step goals. Current approaches rely on environment-agnostic imitation learning that disconnects instructions from environmental contexts, causing models to struggle with context-sensitive instructions and rely on supplementary cues rather than visual reasoning during long-horizon interactions. In this work, we propose **W**orld-**A**ware **P**lanning Narrative Enhancement (**WAP**), a framework that infuses LVLMs with comprehensive environmental understanding through four cognitive capabilities (visual appearance modeling, spatial reasoning, functional abstraction, and syntactic grounding) while developing and evaluating models using only raw visual observations through curriculum learning. Evaluations on the EB-ALFRED benchmark demonstrate substantial improvements, with Qwen2.5-VL achieving a 60.7 absolute improvement in task success rates—particularly in commonsense reasoning (+60.0) and long-horizon planning (+70.0). Notably, our enhanced open-source models outperform proprietary systems like GPT-4o and Claude-3.5-Sonnet by a large margin.

## 1 Introduction

Recent advances in Large Vision-Language Models (LVLMs) [6, 20] have expanded their applications to embodied planning tasks, where agents interpret natural language instructions into a sequence of actions which will be further executed in interactive environments [21, 28]. These models leverage large-scale pretraining on vision-language datasets to align visual inputs with textual commands, achieving notable success in controlled scenarios with explicit object references and low environmental complexity [13, 12]. However, when faced with increasingly complex real-life-like scenarios—those involving unfamiliar environments, varied instruction formats, and multi-step goals—current methods exhibit severe limitations in both generalization ability and reasoning consistency[26].

A fundamental challenge confronting current embodied planning systems lies in their environment-agnostic imitation learning paradigm. In existing methodologies [13, 12], expert demonstration trajectories are typically associated with simplified, environment-independent instructions (e.g., "put the apple on the table"). They force models to learn direct mappings from generic instructions to action sequences in an open-loop manner, disregarding the nuances of the changing surrounding environment. This learning approach operates on a disjointed fashion, treating task instructions and environmental

---

\*Equal contribution

†Corresponding author

contexts as distinct, unconnected elements. This hinders models from developing an integrated grasp of the environment's specific characteristics such as visual appearance, spatial relationships, object functionality, and linguistic comprehension—capabilities essential for human-like task execution [9]. While these models may perform adequately in standardized validation environments, their performance significantly deteriorates when faced with difficult situations that demand the ability to establish links between the changing surroundings and context-sensitive instructions, for example, "place the apple on the table near the television". The limitation is becomes evident during extended sequences of interaction, where models, due to their lack of detailed environmental representations, struggle to integrate previous visual observations. Hence, they resort to supplementary environmental cues, like feedback from actions taken or indicators of task progress, instead of relying exclusively on visual input for decision-making.

To address these gaps, we introduce WAP, an innovative world-aware narrative enhancement approach. This method is designed to infuse LVLMs with comprehensive information about the environment. By doing so, we aim to augment the model's understanding of the environment by incorporating relevant context from the world around it. Inspired by traditional theories of cognitive intelligence [14], our narrative enhancement focuses on collecting data that progressively cultivates four interconnected capabilities: (1) *Visual Appearance Modeling*: Detailed capture of object textures and geometries. (2) *Spatial-Relational Reasoning*: Understanding the spatial arrangement and room layouts. (3) *Functional Abstraction Learning*: Grasping tool-object relationships and symbolic representations. (4) *Syntactic Grounding*: Interpreting complex language to resolve ambiguity. To effectively steer the data generation process while considering various dimensions, we augment existing disjointed data by integrating complementary information from the environment or semantic spaces. Secondly, our framework adheres to realistic deployment scenarios: agents receive only image observations and natural language instructions in a closed-loop manner, without auxiliary privileged environmental feedback. To incrementally equip the model to tackle cognitive challenges, we collect the data and train the model with a curriculum learning strategy, which enhances the model's ability to formulate sophisticated strategies in a wide range of situations, markedly differentiating from traditional imitation learning techniques that do not support this level of advanced intellectual growth.

We evaluate our framework through extensive experiments using Qwen2.5-VL [8]and InternVL3 [32] on the EB-ALFRED benchmark within EmbodiedBench [26], a challenging suite of high-level planning tasks. Our approach achieves substantial improvements over baseline methods, with Qwen2.5-VL demonstrating a 60.7 absolute improvement in average task success rates. Particularly noteworthy are the significant gains in commonsense reasoning (+60.0) and long-horizon planning (+70.0), with similar patterns observed for InternVL3. Remarkably, our enhanced open-source models outperform recent proprietary systems like GPT-4o and Claude-3.5-Sonnet by a large margin.

This work makes three key contributions:

- Our approach bridges the gap between high-level task instructions and the nuanced details of real-world environments by integrating contextual world knowledge into planning systems. This multidimensional enhancement leverages narratives that are aware of various environmental factors, making the planning process more robust and adaptable to complex, real-life scenarios.

- We demonstrate that closed-loop embodied agents can achieve superior planning performance using only visual observations and natural language instructions without privileged environmental feedback—challenging the prevailing assumption that additional auxiliary signals are necessary for robust planning in complex environments.

- Our approach establishes new state-of-the-art benchmarks on EB-ALFRED, outperforming not only existing academic baselines by 60.7 improvement but also surpassing proprietary systems like GPT-4o and Claude-3.5-Sonnet by significant margins in challenging long-horizon planning scenarios.

## 2 Related Works

Embodied planning [4, 25] represents a critical cognitive capability for embodied agents, serving as the brain that guides physical interactions within complex environments. Early approaches to embodied planning [18, 27, 31] relied exclusively on textual environment metadata rather than visual perception, limiting their adaptability to real-world scenarios. Later visual-based methods

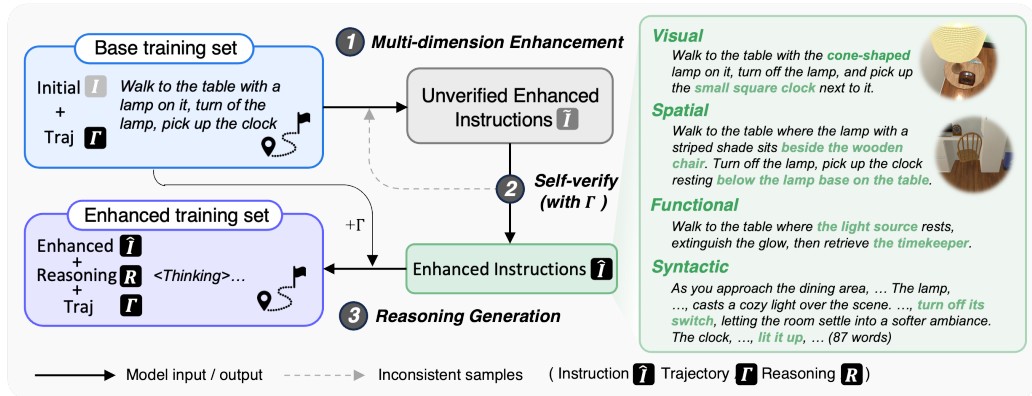

Figure 1: **Multi-dimensional cognitive enhancement framework overview.** Our approach transforms base instruction-trajectory pairs through a structured pipeline: (1) Four-dimensional instruction augmentation, targeting different cognitive abilities; (2) Reasoning generation with semantic verification; and (3) Curriculum learning across progressive stages. This process builds robust planning capabilities while maintaining strict visual-only observation constraints.

[7, 11, 15, 16, 30] employed cascaded pipelines with external models for visual processing like semantic maps. Recent LVLM-based systems [21, 25] have begun processing visual input directly but continue to depend on unrealistic forms of environmental feedback like action success signals and task progress information that would be unavailable in genuine deployment scenarios. Our work advances the field by operating solely on raw visual observations without privileged feedback and by processing complete observational history, more closely mirroring human capabilities in real-world settings.

Methodologically, embodied planning approaches can be categorized as either training-free or training-based. Training-free methods leverage prompt engineering [16, 10, 3] or multi-agent frameworks with specialized roles [29, 5, 22]. Training-based approaches include supervised fine-tuning on human expert demonstrations [23, 2], or methods that recognize the importance of trial-and-error learning through either direct preference optimization [17, 30, 21] or reinforcement learning [1, 24, 19]. Unlike these approaches that often treat task instructions and environmental contexts as disconnected elements, our world-aware planning narrative enhancement framework systematically develops integrated cognitive capabilities, enabling LVLMs to form sophisticated planning strategies without relying on privileged cues, more closely resembling human cognitive processes in complex real-world scenarios.

## 3 Methodology

We propose a multi-dimensional cognitive enhancement framework for embodied planning that systematically develops reasoning capabilities across complementary dimensions while operating solely on raw visual observations. Figure 1 illustrates our approach's core components.

### 3.1 Problem Formulation

In embodied planning tasks, an agent must execute a sequence of actions $\{a_1, a_2, \ldots, a_T\}$ based on a natural language instruction $I$ and egocentric visual observations $\{o_1, o_2, \ldots, o_T\}$. We operate within a closed-loop control framework, where each action decision depends critically on both current observation and historical context. This mirrors real-world robotics scenarios where agents must continuously adapt to environmental changes resulting from previous interactions. Unlike many previous approaches that focus on isolated decision points, our framework explicitly models temporal dependencies by conditioning each action on the complete observation history:

$$a_t = f_\Theta(I, \{o_1, o_2, \ldots, o_t\}) \tag{1}$$

Where $f_\Theta$ represents our vision-language model that predicts the next action based on the instruction and observation history. The key challenge is to develop robust planning capabilities across diverse scenarios with only visual-context history. Our multi-dimensional cognitive enhancement approach specifically addresses these challenges by systematically developing the agent's ability to extract and integrate relevant information across time.

## 3.2 Multi-Dimensional Cognitive Enhancement

Our complete narratives enhance framework follows the pipeline illustrated in Figure **??**. Our framework enhances cognitive capabilities across four complementary dimensions that systematically develop different aspects of the embodied intelligence required for robust planning.

### 3.2.1 Instruction Augmentation

Given an original instruction $I$ and expert trajectory $\tau = \{a_t\}_{t=1}^{T}$ with corresponding observations $\{o_t\}_{t=1}^{T}$, we generate enhanced instructions $\{\tilde{I}_k\}$ across four cognitive dimensions:

$$\tilde{I}_k = \mathcal{M}(I, o_T; \theta_k), \quad k \in \{\text{Visual}, \text{Spatial}, \text{Functional}, \text{Syntactic}\} \tag{2}$$

Where $\mathcal{M}$ is a superior vision-language model and $\theta_k$ represents dimension-specific prompting strategies:

- **Visual Dimension:** Enhances object appearance modeling by explicitly describing visual attributes critical for identification (e.g., *cone-shaped lamp*, *small square clock*).
- **Spatial Dimension:** Augments positional understanding by specifying object locations relative to environmental landmarks (*beside the wooden chair*) and precise spatial relationships (*below the lamp base*).
- **Functional Dimension:** Develops deeper object-interaction understanding by articulating affordances and functional properties (*light source*, *timekeeper*) that capture causal relationships between objects.
- **Syntactic Dimension:** Introduces linguistic complexity through narrative structures, indirect references, and contextual dependencies that require resolving ambiguity beyond literal instructions.

This augmentation process creates a structured hierarchy of cognitive demands, progressively challenging the model's environmental reasoning capabilities.

### 3.2.2 Semantic Consistency Verification

To ensure augmented instructions maintain task equivalence, we implement a verification mechanism:

$$\mathcal{C}(\tilde{I}) = 4 \leq \sum_{i=1}^{5} \mathbb{I}\left( \mathcal{V}(I, \tilde{I}, \{o_i : i \in \{1 \cdots T\}\}; \phi_i) = 1 \right) \tag{3}$$

Where $\mathcal{V}(\cdot; \phi_i)$ represents the verification function that checks if the generated instruction $\tilde{I}$ represents the same intention as $I$, $\mathbb{I}(\cdot)$ is an identification function and evaluates to 1 if the input expression is true. Instructions fail to meet the threshold will trigger a regeneration of the instruction to maintain dataset quality.

### 3.2.3 Stepwise Reasoning Generation

For each action $a_t$ in trajectory $\tau$, we generate explicit reasoning $r_t$ that captures the cognitive process linking observation to action:

$$r_t = \mathcal{M}(\tilde{I}, o_t, \{(\cdot, a_t)\} \cup \{(r_i, a_i) : i \in \{1, \cdots, t-1\}\}) \tag{4}$$

The $t$-th step of reasoning is left out for the vision-language model $\mathcal{M}$ to predict. These reasoning annotations serve as intermediate supervision signals that help the model develop explicit cognitive processes that might otherwise remain implicit, including environmental state tracking, object relationship inferences, and action preconditions.

### 3.3 Curriculum Learning Framework

Our training procedure follows a three-stage curriculum that gradually increases cognitive complexity:

$$\Theta^* = \arg\min_{\Theta} \sum_{s=1}^{3} \mathbb{E}_{(\tau, I) \sim \mathcal{D}_s} \mathcal{L}_{\text{CE}}(f_{\Theta}(\{o_{1:t}\}, I), a_t) \tag{5}$$

where $f_{\Theta}$ represents the vision-language model being optimized and $\mathcal{L}_{\text{CE}}$ is the cross-entropy loss for action prediction. The curriculum stages are:

1. **Base Stage** ($\mathcal{D}_1$): Training on original instruction-trajectory pairs to establish foundational action mapping capabilities.

2. **Environmental Understanding Stage** ($\mathcal{D}_2$): Incorporating visual and spatial augmentations to develop perceptual grounding and scene comprehension.

3. **Conceptual Reasoning Stage** ($\mathcal{D}_3$): Introducing functional and syntactic augmentations to develop higher-order reasoning about object relations and ambiguous references.

This progressive training scheme aligns with cognitive development theories, allowing the model to first master perception-action correspondences before tackling more abstract semantic relationships. Importantly, our framework operates under strict partial observability constraints—providing only egocentric RGB observations without privileged information—to ensure real-world applicability.

## 4 Experiments

We conduct comprehensive experiments to evaluate our framework's effectiveness in enhancing embodied planning capabilities. Our analysis focuses on both overall performance metrics and fine-grained cognitive capabilities across diverse task contexts.

### 4.1 Experimental Settings

**Dataset** We construct an enhanced corpus comprising 80,875 instruction-trajectory pairs derived from the original 16,145 ALFRED trajectories through our multi-dimensional augmentation approach. The dataset is structured across four cognitive dimensions: Visual (appearance attributes), Spatial (positional relationships), Functional (interaction affordances), and Syntactic (referential complexity). This structured enhancement enables systematic evaluation of specific cognitive capabilities crucial for embodied agents.

**Models** For instruction augmentation and reasoning generation described in Section 3, we employ Qwen2.5-VL-72B-Instruct as the teacher model. We evaluate our framework on two foundation model series: Qwen2.5-VL (Qwen2.5-VL-7B-Instruct) [8] and InternVL3 (InternVL3-8B) [32], representing state-of-the-art vision-language architectures with distinct pretraining approaches.

**Evaluation** We evaluate on the EB-ALFRED benchmark from EmbodiedBench [25], which provides refined evaluation protocols over the original ALFRED benchmark, including streamlined action spaces and higher-quality language instructions. Beyond the standard Success Rate (SR) metric, we also use the standard deviation to quantify a model's ability to maintain consistent performance across varying task complexities:

$$\text{STD} = \sqrt{\frac{1}{6} \sum_{c \in \mathcal{C}} (SR_c - \overline{SR})^2} \tag{6}$$

Where $\mathcal{C}$ represents the six task categories (Base, Common, Complex, Visual, Spatial, and Long), and $\overline{SR}$ is the mean success rate across all categories. Lower STD values indicate more balanced capabilities across different task types, while higher values suggest uneven performance that excels in some categories but struggles in others, representing lower robustness. This metric provides crucial insight into model robustness beyond aggregate performance measures.

Table 1: Performance comparison on EmbodiedBench (EB-ALFRED). Results show success rates (SR) across task categories. Models marked with † indicate our proposed approach and its variants. Results for InternVL3-8B and harder closed-loop settings are conducted by ourselves. Best results in **bold**.

| Model | Avg. | STD↓ | Base | Common | Complex | Visual | Spatial | Long |
|---|---|---|---|---|---|---|---|---|
| *Proprietary Models (Original open-loop setting with action feedback)* | | | | | | | | |
| GPT-4o | 56.3 | 7.8 | 64 | 54 | 68 | 46 | 52 | 54 |
| Claude-3.5-Sonnet | 64.0 | 8.6 | 72 | 66 | 76 | 60 | 58 | 52 |
| Gemini-1.5-Pro | 62.3 | 7.8 | 70 | 64 | 72 | 58 | 52 | 58 |
| Gemini-2.0-flash | 52.3 | 6.2 | 62 | 48 | 54 | 46 | 46 | 58 |
| Gemini-1.5-flash | 39.3 | 10.6 | 44 | 40 | 56 | 42 | 26 | 28 |
| GPT-4o mini | 24.0 | 13.0 | 34 | 28 | 36 | 24 | 22 | 0 |
| *Open-Source Models* | | | | | | | | |
| InternVL2.5-78B-MPO | 40.0 | 4.5 | 48 | 36 | 42 | 40 | 40 | 34 |
| Qwen2.5-VL-72B-Ins | 39.7 | 6.3 | 50 | 42 | 42 | 36 | 34 | 34 |
| Qwen2-VL-72B-Ins | 33.7 | 4.8 | 40 | 30 | 40 | 30 | 32 | 30 |
| Llama-3.2-90B-Vision-Ins | 32.0 | 10.1 | 38 | 34 | 44 | 28 | 32 | 16 |
| InternVL2.5-38B-MPO | 25.7 | 4.7 | 30 | 20 | 20 | 28 | 32 | 24 |
| InternVL2.5-38B | 23.3 | 9.0 | 36 | 30 | 36 | 22 | 14 | 26 |
| Llama-3.2-11B-Vision-Ins | 13.7 | 7.4 | 24 | 8 | 16 | 22 | 6 | 6 |
| InternVL2.5-8B-MPO | 7.7 | 4.3 | 12 | 6 | 14 | 6 | 6 | 2 |
| Qwen2.5-VL-7B-Ins | 4.7 | 3.9 | 10 | 8 | 6 | 2 | 0 | 2 |
| Qwen2-VL-7B-Ins | 1.7 | 2.3 | 6 | 0 | 2 | 0 | 0 | 2 |
| InternVL3-8B | 10.7 | 7.6 | 20 | 12 | 20 | 8 | 2 | 2 |
| *Proprietary Models (Under harder closed-loop settings)†* | | | | | | | | |
| GPT-4o | 26 | 2.8 | 30 | 26 | 28 | 22 | 26 | 24 |
| Claude-3.5-Sonnet | 57.3 | 13.4 | 62 | **62** | 64 | 40 | 42 | **74** |
| *Our Approach with InternVL3-8B†* | | | | | | | | |
| InternVL3-8B | 6 | 4.7 | 12 | 8 | 10 | 2 | 0 | 4 |
| + Original Reasoning | 46.0 | 18.4 | 58 | 16 | 56 | 46 | 34 | 66 |
| + WAP Augmentation | 57.0 | 5.5 | 62 | 52 | 60 | 58 | 48 | 62 |
| + Curriculum Learning | 61.0 | 7.2 | 66 | 56 | 66 | 58 | 50 | 70 |
| *Our Approach with Qwen2.5-VL-7B-Ins†* | | | | | | | | |
| Qwen2.5-VL-7B-Ins | 2 | 2.5 | 6 | 2 | 4 | 0 | 0 | 0 |
| + Original Reasoning | 47.0 | 14.0 | 64 | 22 | 48 | 50 | 44 | 54 |
| + WAP Augmentation | 58.0 | 6.8 | 60 | 62 | 62 | 46 | 54 | 64 |
| + Curriculum Learning | **62.7** | 6.3 | **66** | **62** | **70** | **56** | **52** | 70 |

## 4.2 Main results

As demonstrated in Table 1, our curriculum learning framework establishes new state-of-the-art performance across all task categories while operating under strictly realistic observation constraints. The Qwen2.5-VL implementation achieves a 13.5× improvement in average success rate (from 4.67 to 62.67) with 14% improvement compared with baseline method, surpassing even GPT-4o (56.3) and approaching Gemini-1.5-Pro (62.3) under harder settings — without having access to privileged environmental information. Similar performance gains are observed with InternVL3 models, which improve from 10.67 to 61.0.

Notably, our approach maintains a competitive Standard Deviation (STD) of 6.3, lower than many proprietary models such as Claude-3.5-Sonnet (8.6) and Gemini-1.5-flash (10.6), indicating more balanced capabilities across diverse task categories. This represents a substantial improvement over our basic reasoning approach, which exhibits a high STD of 14.0, suggesting uneven performance that handles some categories well but struggles significantly with others. The progression from basic reasoning (STD=14.0) to curriculum learning (STD=6.3) demonstrates how our multi-dimensional enhancement approach systematically builds more balanced cognitive capabilities.

**Enhanced Environmental Cognition** Our framework demonstrates substantial improvements in environmental understanding capabilities, evidenced by performance across specialized task categories:

1. **Visual Perception:** For InternVL3, the success rate improves from 46 (baseline) to 58 on tasks requiring the identification of objects based on appearance (e.g., distinguishing objects by color, shape, or pattern).

2. **Spatial Reasoning:** Performance rises from 34 to 50 (InternVL3) on tasks requiring precise positioning and relational understanding (e.g., "place the object to the left of the sink").

3. **Semantic Grounding:** For Qwen2.5-VL, commonsense reasoning accuracy increases from 22 to 62, enabling effective handling of instructions requiring real-world knowledge and functional understanding of objects.

4. **Referential Resolution:** The model successfully resolves ambiguous references (e.g., "that container mentioned before") through contextual reasoning with accuracy raising from 48 to 70 (Qwen2.5-VL), overcoming a critical limitation in conventional vision-language systems.

These improvements underscore the importance of structured environmental modeling beyond simple action prediction for robust embodied planning.

**Further generalization on long-horizon planning** Our approach achieves remarkable 70 success rate on long-horizon tasks–those requiring 15+ sequential actions–representing a 35-fold improvement over baseline models and matching Claude-3.5-Sonnet's performance in this challenging category. Notably, while proprietary models show mixed results under the stricter closed-loop setting (without action feedback), our framework maintains consistent performance across all settings.

This exceptional capability stems from two key innovations, (1) **Full Temporal Context:** Our training paradigm incorporates complete observation histories rather than isolated frames, enabling the model to capture causal relationships between actions and environmental changes across extended sequences. (2) **Multi-dimensional Knowledge Integration:** The four cognitive dimensions of our data enhancement approach collectively enable the model to maintain coherent world representations throughout extended task execution.

The dramatic performance disparity between GPT-4o (24 in closed-loop vs. 54 in open-loop settings) and Claude-3.5-Sonnet (74 in closed-loop vs. 52 in open-loop) on long-horizon tasks is particularly revealing. While GPT-4o struggles without action feedback, likely falling into repetitive error patterns, Claude-3.5-Sonnet actually improves under closed-loop constraints, suggesting superior error recognition and recovery capabilities. Our framework (70 success rate) approaches Claude's closed-loop performance without requiring proprietary model access, validating our hypothesis that proper environmental modeling serves as a critical foundation for compositional task planning. This confirms that closed-loop settings, despite being more challenging, better reflect the requirements for successful long-horizon planning in real-world scenarios.

## 5 Analysis

### 5.1 Self-Directed Enhancement Potential

To explore autonomous data augmentation capabilities, we investigate a self-directed enhancement approach where models independently select augmentation strategies based on task descriptions. This implicit enhancement contrasts with our explicit curriculum learning framework.

As shown in Table 2, self-directed enhancement achieves moderate success (56.7 average), but exhibits notable limitations compared to our explicit framework (62.7). The self-directed approach demonstrates reasonable environmental perception capabilities—matching explicit methods in Visual (52 vs. 56) and Spatial (52 vs. 52) categories. However, it shows significant deficiencies in semantic reasoning: Commonsense understanding (48 vs. 62) and Long-horizon planning (60 vs. 70).

While the self-directed curriculum approach improves performance consistency (STD=7.2) compared to the basic reasoning baseline (STD=14.0), it remains less balanced than our explicit curriculum method (STD=6.3). This suggests that while the model can autonomously develop more uniform capabilities across tasks, it still benefits from structured guidance in developing complementary

Table 2: Comparison between explicit and self-directed enhancement approaches

| Method | Avg | STD↓ | Base | Common | Complex | Visual | Spatial | Long |
|---|---|---|---|---|---|---|---|---|
| Original Reasoning | 47.0 | 14.0 | 64 | 22 | 48 | 50 | 44 | 54 |
| Explicit (Standard) | 58.0 | 6.8 | 60 | 62 | 62 | 46 | 54 | 64 |
| Explicit (Curriculum) | **62.7** | **6.3** | **66** | **62** | **70** | **56** | **52** | **70** |
| Self-Directed (Standard) | 54.7 | 7.7 | 60 | 50 | 62 | 44 | 50 | 62 |
| Self-Directed (Curriculum) | 56.7 | 7.2 | 66 | 48 | 62 | 52 | 52 | 60 |

cognitive skills. The model shows particular weakness in commonsense reasoning (48), indicating a tendency toward surface-level linguistic modifications rather than deeper semantic understanding.

Nevertheless, the 56.7 success rate achieved without human-designed enhancement rules suggests promising avenues for autonomous improvement in future work.

## 5.2 Generalization to Unseen Environments and Object Configurations

To evaluate robustness beyond the training distribution, we measured performance on the unseen split of VOTA-Bench [21], which introduces novel kitchens, living rooms, and object instances absent during training. We report Success Rate (SR) and path-length–weighted success (PL) across diverse tasks, comparing our approach (WAP) with GPT-4o:

Table 3: Results on VOTA-Bench unseen split. SR = success rate; PL = path-length–weighted success.

| Task | Examine & Light | | Pick & Place | | Stack & Place | | Heat & Place | | Cool & Place | | Overall | |
|---|---|---|---|---|---|---|---|---|---|---|---|---|
| Method | SR | PL | SR | PL | SR | PL | SR | PL | SR | PL | SR | PL |
| GPT-4o | 30.42 | 24.73 | 24.15 | 19.82 | 18.27 | 14.56 | 16.83 | 15.98 | 12.54 | 10.45 | 20.36 | 16.83 |
| WAP (ours) | 70.35 | 65.42 | 58.62 | 54.91 | 61.47 | 56.37 | 63.84 | 60.21 | 59.28 | 57.63 | 64.56 | 59.86 |

WAP consistently outperforms GPT-4o across all task families and both metrics, indicating improved grounding and execution efficiency in environments and object configurations that differ substantially from those seen during training.

## 5.3 Sensitivity to Teacher Model Quality and Size

We assess sensitivity to the teacher model by varying size and quality using Qwen2.5-7B, Qwen2.5-32B, and Qwen2.5-72B [8]. Larger teachers generally yield better results across most dimensions, with stronger gains in spatial and long-horizon reasoning:

Table 4: Impact of teacher model capacity on performance across evaluation aspects.

| Teacher | Avg | Base | Visual | Spatial | Common | Complex | Long |
|---|---|---|---|---|---|---|---|
| Qwen2.5-7B | 54.67 | 66 | 54 | 56 | 48 | 48 | 56 |
| Qwen2.5-32B | 57.33 | 64 | 60 | 62 | 52 | 46 | 60 |
| Qwen2.5-72B | 62.67 | 66 | 62 | 70 | 56 | 52 | 70 |

As shown in Table 4, average performance improves monotonically (54.67→57.33→62.67), with consistent gains in the Spatial and Long categories as the teacher scales up. Some dimensions (e.g., Complex) may show non-monotonic changes at intermediate scales, suggesting that benefits from additional capacity can be task- and skill-dependent.

## 5.4 Ablation Study

We conduct a top-down ablation on Qwen2.5-VL-7B by starting from the full framework and progressively disabling modules to isolate their effects. We remove, in sequence: (i) curriculum learning (three-stage schedule: Base → Environmental Understanding → Conceptual Reasoning); (ii)

Table 5: Top-down ablation from the full framework by progressively disabling curriculum, step-wise reasoning, and WAP instruction enhancement

| Configuration | Avg | STD↓ | Base | Common | Complex | Visual | Spatial | Long |
|---|---|---|---|---|---|---|---|---|
| Curriculum (Full) | **62.7** | **6.3** | **66** | **62** | **70** | **56** | **52** | **70** |
| w/o Curriculum | 58.0 | 6.8 | 60 | 62 | 62 | 46 | 54 | 64 |
| w/o Following-steps Reasoning | 54.0 | 9.3 | 62 | 46 | 64 | 52 | 40 | 60 |
| w/o First-step Reasoning | 46.7 | 17.1 | 60 | 16 | 56 | 46 | 42 | 60 |
| w/o WAP Instruction | 47.0 | 14.0 | 64 | 22 | 48 | 50 | 44 | 54 |
| Base Model | 4.7 | 3.9 | 10 | 8 | 6 | 2 | 0 | 2 |

step-wise reasoning components—first-step reasoning and the reasoning chains applied at following steps; and (iii) WAP instruction enhancement (multi-dimensional prompts integrating visual, spatial, commonsense, and long-horizon cues).

According to the results in Table 5, removing curriculum decreases the average by 4.7 points (62.7→58.0) and slightly increases variance. The largest drops occur in Visual (56→46, -10), Complex (70→62, -8), Long (70→64, -6), and Base (66→60, -6), while Common remains unchanged (62) and Spatial slightly improves (52→54, +2). This indicates that curriculum primarily benefits perceptual robustness and long-horizon execution, with limited direct impact on commonsense.

Disabling reasoning chains after the first step (w/o Following-steps Reasoning) further reduces the average to 54.0 (-4.0) and increases variance (STD 6.8→9.3). The strongest declines are in Common (62→46, -16) and Spatial (54→40, -14), underscoring that sustained step-wise reasoning is crucial for coherent environment–action coupling beyond initial guidance. Visual and Complex rise slightly (46→52; 62→64), suggesting that front-loaded guidance offers short-term benefits for perceptual tasks but cannot maintain commonsense or spatial consistency.

Removing the initial bootstrapping chain (w/o First-step Reasoning) causes a sharper average drop to 46.7 (-7.3) and the largest variance increase (STD 9.3→17.1). Common collapses (46→16, -30), with additional declines in Complex (64→56, -8) and Visual (52→46, -6), while Long remains unchanged (60). This pattern shows that first-step reasoning is pivotal for establishing correct task setup and commonsense grounding; without it, the model becomes highly unstable across categories.

Replacing WAP prompts with original simple instructions (w/o WAP Instruction) keeps the average similar (46.7→47.0) but changes the error profile: Common recovers modestly (16→22, +6), whereas Complex (56→48, -8) and Long (60→54, -6) deteriorate. This suggests WAP cues are particularly beneficial for complex and long-horizon behavior, but in the absence of step-wise reasoning they can amplify imbalance across categories.

Overall, three findings emerge: (1) Curriculum chiefly strengthens long-horizon planning and perceptual robustness while modestly reducing variance. (2) Complete step-wise reasoning is essential for stability and for commonsense and spatial coherence; removing follow-up chains disproportionately harms Common and Spatial. (3) WAP instruction is necessary to realize gains on Complex and Long tasks, but without sustained reasoning chains it yields an imbalanced skill profile. The full combination achieves the highest average with the lowest variance, improving the base model by over 13× while maintaining balanced gains across all splits.

## 5.5 Case Study

To provide qualitative insights into our model's capabilities, we analyze representative examples of embodied planning and reasoning. Our case studies focus on (1) cognitive process transparency in complex multi-step planning and (2) the effectiveness of our multi-dimensional augmentation approach.

Figure 2 illustrates the execution of a seemingly simple instruction: "Place a chilled apple section into the bin." This instruction contains implicit procedural requirements—the apple must be sliced and chilled—necessitating a sequence of steps beyond merely disposing of an apple.

Our Qwen2.5-VL model with curriculum learning successfully decomposes this task into 18 distinct actions across three phases. First, the model identifies necessary tools, demonstrating task decomposition by recognizing that sectioning requires a knife. Second, it exhibits situational awareness,

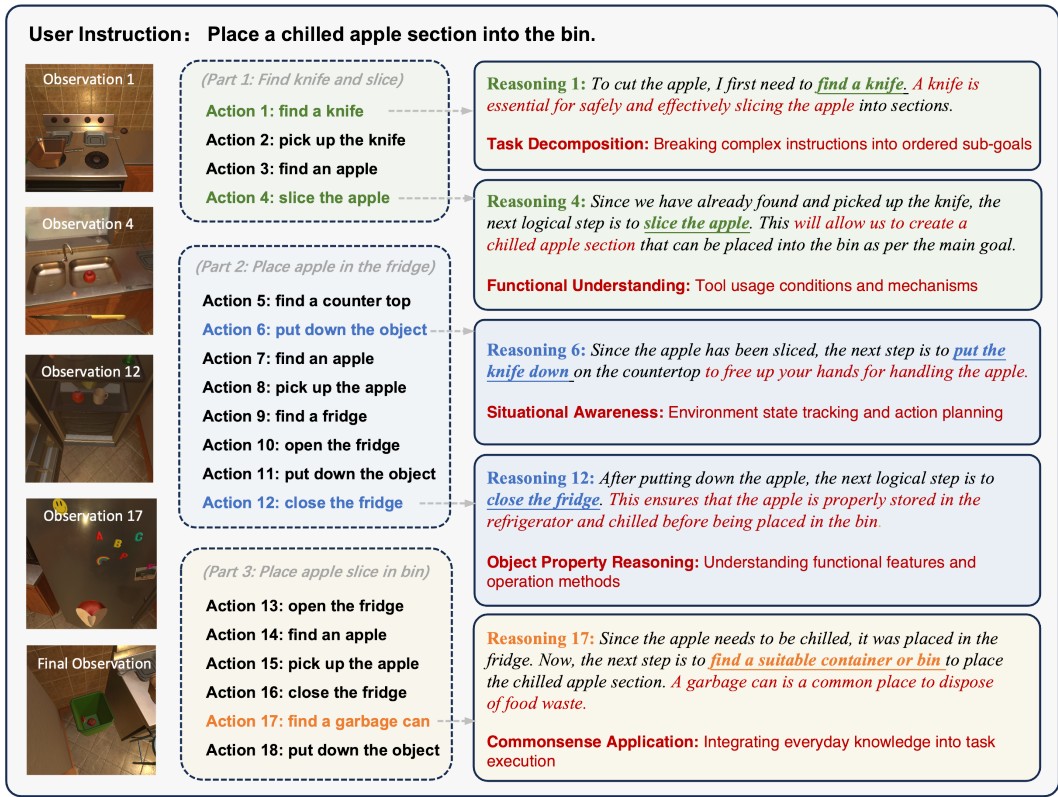

Figure 2: **Reasoning process visualization for complex instruction execution.** The figure shows our model executing the instruction "Place a chilled apple section into the bin." The model successfully decomposes this seemingly simple instruction into 18 distinct actions across three phases, demonstrating robust planning capabilities. Reasoning annotations highlight five critical cognitive abilities: task decomposition, functional understanding, situational awareness, object property reasoning, and commonsense knowledge application. This example illustrates how our model maintains coherent planning over a long horizon (18 steps) while handling implicit requirements not explicitly stated in the instruction (e.g., the apple must be chilled before disposal).

notably in Reasoning 6 where it places the knife down before handling the apple, showing safety awareness. Most importantly, the model correctly infers the need to refrigerate the apple to fulfill the "chilled" requirement—demonstrating both functional understanding of appliances and commonsense knowledge application. This example highlights improvements achieved through our curriculum learning approach, as baseline models consistently failed to maintain coherence across extended sequences, typically omitting the critical chilling step.

## 6 Conclusion

In this paper, we address fundamental limitations in embodied agents where current LVLMs struggle with complex real-life scenarios due to their environment-agnostic imitation learning paradigm. Previous approaches treat task instructions and environmental contexts as disconnected elements, leading to poor generalization in unfamiliar environments and inconsistent reasoning during multi-step tasks. We introduce a world-aware narrative enhancement approach that systematically develops four cognitive capabilities: visual appearance modeling, spatial-relational reasoning, functional abstraction learning, and syntactic grounding. Our experiments on EB-ALFRED demonstrate substantial improvements with Qwen2.5-VL and InternVL3, achieving up to 60.7 higher success rates and outperforming even proprietary models like GPT-4o in challenging scenarios. Our approach proves that high-performance embodied planning is possible using only raw visual observations without privileged feedback, establishing a new state-of-the-art for embodied AI systems in complex, real-life-like environments.

## Acknowledgment

This work was supported by the Science and Technology Commission of Shanghai Municipality (No. 24511103100).

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

# 7 Technical Appendices and Supplementary Material

## A  Limitation

While our framework demonstrates significant advancements in embodied planning, several limitations warrant further discussion. First, our world-aware narrative enhancement operates primarily at the symbolic action level (e.g., "pick up knife"), lacking explicit modeling of continuous control parameters. This abstraction necessitates future integration with low-level controllers to enable practical deployment requiring force modulation and precise trajectory optimization. Second, while our experiments demonstrate effectiveness in household environments based on the ALFRED dataset, the framework's generalizability to industrial settings or outdoor scenarios with dynamic obstacles remains unverified. These domains may present distinct challenges in spatial reasoning and adaptive navigation that our current implementation does not address. Third, in our approach we mainly focus on enhancing user instruction and step-wise reasoning, limiting the fine-tuned model's capacity for mid-execution error correction. This constraint suggests the need for more dynamic enhance mechanisms in future iterations.

## B  Training Details

### B.1  Narrative Enhancement

Our instruction enhancement pipeline employs the Qwen2.5-VL-72B-Instruct model under manufacturer-recommended configurations, utilizing temperature sampling ($\tau$=1.0) with nucleus filtering (top-p=0.7) for balanced diversity and coherence. This process generates both narrative-augmented instructions and associated step-wise rationales, maintaining strict alignment with the original ALFRED action trajectories through constrained decoding mechanisms.

### B.2  Model Fine-tuning

We implement full-parameter optimization on two vision-language architectures: Qwen2.5-VL-72B-Instruct and InternVL3-8B. The training regime employs AdamW optimization with base learning rate $\eta = 1 10^{-5}$, 10% linear warmup, and cosine decay scheduling over 3 epochs. Experiments utilize contrastive context windows (16k/32k tokens) with per-device batch size 4, distributed across 8×A100-80GB nodes via tensor parallelism. The complete training cycle requires 14 hours per model variant, aggregating to 800 A100 GPU-hours when accounting for ablation studies and hyperparameter tuning. Memory optimization is achieved through Flash Attention v2 and BF16 mixed-precision training, maintaining numerical stability while maximizing hardware utilization.

### B.3  Compute for augmentation and training.

We further report the end-to-end compute for data preparation and training on H100-class accelerators:

Table 6: Compute profile for augmentation and training.

| Stage | Hardware | Compute (GPU hours) |
| --- | --- | --- |
| Instruction augmentation | 4× H100 | 20 |
| Reasoning augmentation | 4× H100 | 200 |
| Training | 8× H100 | 100 |

Reasoning augmentation dominates the preparation cost, followed by supervised training and instruction augmentation. In practice, practitioners can: (i) adopt complete observation history at inference for stronger execution performance; and (ii) scale reasoning augmentation selectively to balance up-front cost against downstream improvements.

# C  Additional Results and Analyses

## C.1  Performance–Cost Trade-offs: Observation History and Augmented Instruction Generation

We study how observation history and augmentation choices affect both effectiveness and computational cost. All inference-time results were measured with tensor parallelism on $2\times$ GTX 3090 GPUs. Using complete observation histories (i.e., multi-image inputs) improves performance relative to a single observation, at the expense of higher inference compute. Importantly, the speed comparison here contrasts multi-image inputs against the original long textual prompts used in EmbodiedBench, isolating the impact of richer visual context versus prompt length.

Table 7: Effect of observation history on performance and inference-time cost (tensor parallel on $2\times$ GTX 3090s).

| Setting | Performance | Inference compute (GPU hours) |
|---|---|---|
| Single observation | 52.33 | 1.6 |
| Complete observation history | 62.67 | 4.2 |

Relative to single-observation inference, complete history yields a +10.34 absolute improvement while increasing compute by roughly $2.6\times$ ($1.6\rightarrow4.2$ GPU hours), offering a clear latency–accuracy knob for long-horizon tasks where temporal context is critical.

## C.2  Qualitative Examples of Reasoning Effect

---
**Example 1: Object Recognition and World Knowledge**

**Task:**
After enjoying your morning coffee, move the beverage holder to the nightstand.

**WAP Model (Success):**
Action Sequence: (1) find a mug (2) pick up the mug (3) find a side table (4) put down object

Step 1 Reasoning: "Since the goal is to move the beverage holder to the nightstand, I need to first locate the beverage holder. Assuming it's a mug, I will start by searching for it in the current environment."
Action: find a mug

**Baseline Model (Failure):**
Action Sequence: (1) find a statue (2) pick up the statue (3) find a side table (4) put down object

Step 1 Reasoning: "Since the goal is to move the beverage holder to the nightstand, I first need to locate the statue to determine its current position and plan the next steps accordingly."
Action: find a statue

---

Figure 3: Example showing baseline's failure to understand object-level world knowledge (mug as beverage holder)

---
**Example 2: Multi-Object Manipulation with Physical Constraints**

**Task:**
When finished cooking, place the saucepan with a spatula inside on the kitchen counter.

**WAP Model (Success):**
Action Sequence: (1) find a spatula (2) pick up the spatula (3) find a pan (4) put down the spatula (5) pick up the pan (6) find a counter top (7) put down object

Step 4 Reasoning: "The current visual state shows that the spatula is being held, but the saucepan is still on the table. To place the spatula inside the saucepan as per the user's instruction, I need to put down the spatula temporarily."
Action: put down the spatula

**Baseline Model (Failure):**
Action Sequence: (1) find a spatula (2) pick up the spatula (3) find a fridge (4) open the fridge (5) find a pan (6) pick up the pan (7) close the fridge (8) find a counter top (9) put down object

Step 3 Reasoning: "After picking up the spatula, the next logical step is to look for a saucepan. A saucepan is typically stored in the fridge or pantry."
Action: find a fridge

**Claude-sonnet-3.5 (Failure):**
Action Sequence: (1) find a pan (2) pick up the pan (3) find a spatula (4) pick up the spatula (5) put down the object in hand (6) find a counter top (7) put down the object in hand

Step 3 Reasoning: "Based on the instruction, we need to handle a saucepan and spatula, placing them together on the kitchen counter. The task requires finding and manipulating multiple objects in a specific sequence to achieve the desired arrangement."
---

Figure 4: Example showing baseline and comparison models' incorrect spatial reasoning and failure to handle one-object-at-a-time constraint

The failure cases reveal different types of reasoning deficiencies: Baseline model shows incorrect spatial reasoning by mistaking sauce and saucepan, demonstrating poor understanding of typical kitchen object locations; Claude-sonnet-3.5 fails to understand the task requirement of placing the spatula inside the pan, instead treating them as separate objects to be placed on the counter; Our model correctly recognizes the one-object-at-a-time constraint and strategically puts down the spatula to enable proper manipulation.

## C.3 Contrastive Examples of Narration-Augmented Training

---
**Original Training Sample**

Original Instruction:
Place white coffee mug on white dresser

Action Sequence: (1) find a mug (2) pick up the mug (3) find a dresser (4) put down object

Step 1 Reasoning: "Since the white coffee mug needs to be placed on the white dresser, I need to locate the mug first before proceeding to the next step."
Action: find a mug
---

Figure 5: Original training sample with basic reasoning

```
┌─ Narration-Augmented Training Sample ─────────────────────────────┐
│                                                                    │
│  Enhanced Instruction:                                             │
│  Place the white vessel for enjoying hot beverages atop the surface│
│  where personal treasures rest.                                    │
│                                                                    │
│  Action Sequence: (1) find a mug (2) pick up the mug (3) find a    │
│  dresser (4) put down the mug                                      │
│                                                                    │
│  Step 1 Reasoning: "The user instruction requires placing a white  │
│  vessel (likely a mug) for enjoying hot beverages on a surface     │
│  where personal treasures rest. From the provided image, there is  │
│  no visible mug present. Therefore, the logical next step is to    │
│  find a mug."                                                      │
│  Action: find a mug                                                │
│                                                                    │
└────────────────────────────────────────────────────────────────────┘
```

Figure 6: Narration-augmented sample with semantic bridges ("vessel for enjoying hot beverages" → mug)

The narration in the augmented data explicitly states the semantic bridges that are only implicit in the original demonstrations. Training on pairs that couple the same action with both concise and enhanced reasoning equips the policy with these paraphrase links. Consequently, at evaluation time the WAP model retrieves the correct object and surface ("beverage holder" → mug, "nightstand" → side table), whereas the baseline—lacking such supervision—often selects objects similar to the training data ("statue") while having no semantic relationships.

# D    Data Generation Process

## D.1    Dataset Statistics

Our enhanced dataset comprises 80,875 instruction-trajectory pairs derived from 16,145 original ALFRED trajectories, systematically expanded through two complementary approaches. The first expansion preserves the original trajectory structure while adding four specialized narrative enhancements (visual, spatial, functional and syntactic), yielding four distinct subsets each containing 16,145 samples. Unlike conventional datasets that provide only sparse instructions and atomic action sequences, our framework enriches each trajectory with: (1) step-wise observation images capturing environmental states, and (2) step-wise reasoning annotations detailing action rationales and preconditions.

Additionally, we provide a comparative dataset of 32,290 samples featuring implicit-instruction augmentation. This contrastive set employs self-supervised prompting techniques where models autonomously determine enhancement requirements through preliminary attention patterns, rather than receiving explicit annotation guidelines. This dual-structure design enables systematic evaluation of both human-guided and model-induced enhancement strategies, while maintaining parity in environmental complexity and task diversity with the original ALFRED distribution.

## D.2    Prompt Templates for Instruction Augmentation

Here are the prompt templates used for generating the enhanced instructions across the four cognitive dimensions.

**Visual Dimension Prompt**

Visual Dimension Prompt:

Enhance the following instruction by adding spatial descriptions based on the image. Keep it natural and concise.

Examples:
1. Original: "Put two spray bottles in the cabinet"
Enhanced: "Place two cylindrical green cleaning spray bottles in the wooden cabinet."

2. Original: "Put a knife in a container"
Enhanced: "Put a 20cm silver chef knife into the blue rectangular plastic container."

Now enhance:
Original: {human_instruction}
Enhanced:

Figure 7: Prompt for instruction visual enhancement

**Spatial Dimension Prompt**

Spatial Dimension Prompt

Enhance the following instruction by adding multi-layered spatial descriptions based on the image. Refer to objects through their positional relationships with 2-3 adjacent landmarks. Keep descriptions natural and concise.

Examples:
1. Original: "Put two spray bottles in the cabinet"
Enhanced: "Put two spray bottles in the white cabinet under the stainless steel sink against the wall"

2. Original: "Put a knife in a container"
Enhanced: "Place the chef's knife holder on the granite countertop, positioned to the right of the refrigerator"

Now enhance:
Original: {human_instruction}
Enhanced:

Figure 8: Prompt for instruction spatial enhancement

**Functional Dimension Prompt**

---

Functional Dimension Prompt

Enhance the following instruction by adding functional descriptions based on world knowledge. Replace one object or its placement with an indirect reference while keeping the sentence natural and concise.

Examples:
1. Original: "Put two spray bottles in the cabinet"
Enhanced: "Place two items used for misting surfaces inside the cabinet."

2. Original: "Put a knife in a container"
Enhanced: "Insert an object commonly used for cutting into a container designed for safekeeping."

Now enhance:
Original: {human_instruction}
Enhanced:

---

Figure 9: Prompt for instruction functional enhancement

**Syntactic Dimension Prompt**

---

Syntactic Dimension Prompt

Enhance the following instruction into a more elaborate version by adding contextual details and symbolic substitutions. Replace objects and locations with contextual references (e.g., pronouns or implied terms) and include irrelevant but plausible background information. Keep sentences concise and avoid adding new actions.

Examples:

1. Original: "Put two spray bottles in the cabinet"
Enhanced: "The spray bottles are on the shelf, already cleaned from yesterday's use. Move them to the cabinet for storage."

2. Original: "Put a knife in a container"
Enhanced: "That knife on the counter just finished slicing vegetables. Place it in the container to keep the edge protected."

3. Original: "Put washed lettuce in the refrigerator"
Enhanced: "There's a lettuce in the sink—we've prepped enough for dinner. Wash it and store there to keep it fresh."

Now enhance:
Original: {human_instruction}
Enhanced:

---

Figure 10: Prompt for instruction syntactic enhancement

### D.2.1 Verification Prompt for Semantic Consistency

> **Verification Prompt for Semantic Consistency**
>
> You are tasked with verifying two instructions describe the same task.
>
> Original instruction: {original_instruction}
> Enhanced instruction: {enhanced_instruction}
> Environment images: [IMAGES]
>
> Please verify if these instructions describe the SAME task goal, even if expressed differently. Consider only task objects and actions, not the specific methods. Respond with ONLY "Yes" if they describe the same task, or "No" if they describe different tasks.

Figure 11: Verification Prompt for Semantic Consist

### D.2.2 Reasoning Generation Prompt

> **Reasoning Generation Prompt**
>
> You are tasked with generating the reasoning process for an embodied agent executing a specific action.
>
> Instruction: {enhanced_instruction}
> Current observation: [IMAGE]
> Current action: {action}
> Previous actions and reasoning:
> {previous_actions_and_reasoning}
>
> Please generate detailed reasoning that explains WHY the agent should take the current action. Include:
> 1. What the agent observes in the environment
> 2. How this relates to the instruction
> 3. Why this specific action is appropriate at this step
> 4. How this action contributes to the overall task goal
>
> Reasoning:

Figure 12: Reasoning Generation Prompt

