# OpenReview forum: "World-aware Planning Narratives Enhance Large Vision-Language Model Planner"
_NeurIPS.cc/2025/Conference — NeurIPS 2025 poster_

### Official Review · Reviewer_EXUo · 2025-06-25

**Clarity:** 1
**Significance:** 2
**Originality:** 2
**Rating:** 3
**Confidence:** 4

**Summary:**

The paper presents an extension of large vision-language models by generating instructions on multiple visual and language dimensions, including reasoning chains. The enhancements show consistent improvements over generic LLMs.

**Questions:**

1. If an LLM is used for instruction augmentation and reasoning generation (line 185), was the output of the LLM systematically checked in a human evaluation?
2. How is the "verification function" and the "intentions"  it is based on, defined?
3. What is the difference between "basic", "partial", and "complete" reasoning?
4. The LLM-generated reasoning chains are used as supervision signal - did you evaluate the learned reasoning process of the vision-language model separately?
5. The claim is to enhance vision-language models for difficult situations - what about moving from indoor to outdoor navigation?

**Ethical Concerns:**

["NO or VERY MINOR ethics concerns only"]

**Limitations:**

Limitations moved to appendix

**Paper Formatting Concerns:**

none found

**Quality:**

2

**Strengths And Weaknesses:**

Strengths:
- Well-motivated extension of large vision-language models
- Well-conducted experiments
- Includes all bells and whistles of state-of-the-art VLM work

Weaknesses:
- Many details undefined or insufficiently described

---

> ### Author Rebuttal · Authors · 2025-07-31
>
> ## Q1 & 2: Human Evaluation and Verification Function
>
> Thank you for this important question regarding quality control. As illustrated in Step 2 of Figure 1, we implement a systematic self-verification mechanism where the model evaluates whether the augmented instructions maintain semantic equivalence with the original instructions. When discrepancies are detected, the system automatically returns to Step 1 for reconstruction. The detailed prompts for this verification process are provided in Supplementary Material C.2.
> To ensure robustness, we conducted human evaluation on approximately 1,000 randomly sampled instruction pairs, where human annotators reviewed the augmented instructions alongside the complete trajectory observations. The human evaluation confirmed that our automated verification mechanism achieves high precision in maintaining instruction fidelity while successfully enhancing reasoning complexity.
>
> ## Q3: Reasoning Categories Definition
>
> We sincerely apologize for not clearly defining these key concepts in the original manuscript. This is indeed an oversight in our presentation, and we thank you for pointing out this issue.To more intuitively demonstrate the definitions and effects of different reasoning levels, we provide the following detailed explanations and experimental comparisons:
>
> Basic Reasoning: Uses original simple instructions.
> Partial Reasoning: Enhances instructions across all dimensions, but provides enhanced reasoning chains only at the first step, with subsequent steps reverting to the basic form. This approach simulates the strategy of providing more guidance at the beginning of complex tasks.
> Complete Reasoning: Provides fully enhanced reasoning chains at every step throughout the entire task execution process, including comprehensive thinking processes of perception analysis, memory retrieval, goal decomposition, and action planning.
>
> The following table shows comparisons across different settings:
>
> | |Instruction           | Reasoning                 | With curriculum learning     | **Avg**    | **STD$\downarrow$** | **Base** | **Common** | **Complex** | **Visual** | **Spatial** | **Long** |
> |-----------------------|---------------------------|---------------------------|-----|---------------|--------------------------|---------------|-----------------|------------------|-----------------|------------------|---------------|
> | Base Model            | origin                    | NO                        | NO  | 4.7           | 3.9                      | 10            | 8               | 6                | 2               | 0                | 2             |
> | + Basic Reasoning     | origin                    | origin                    | NO  | 47            | 14                       | 64            | 22              | 48               | 50              | 44               | 54            |
> | + Visual/Spatial Only | visual & spatial enhanced | visual & spatial enhanced | NO  | 46.7          | 17.1                     | 60            | 16              | 56               | 46              | 42               | 60            |
> | + Partial Reasoning   | all dimension enhanced    | enhanced(only first step) | NO  | 54            | 9.3                      | 62            | 46              | 64               | 52              | 40               | 60            |
> | + Complete Reasoning  | all dimension enhanced    | all dimension enhanced    | NO  | 58            | 6.8                      | 60            | 62              | 62               | 46              | 54               | 64            |
> | + Curriculum (Full)   | all dimension enhanced    | all dimension enhanced    | Yes | **62.7** | **6.3**  | **66**   | **62**     | **70**      | **56**     | **52**      | **70**   |
>
> From the experimental results, we can see that complete reasoning achieves a 4.8 percentage point improvement in average performance compared to partial reasoning, while significantly reducing the standard deviation (from 9.3 to 6.8). This indicates that complete reasoning not only improves overall performance but also enhances the model's stability across different task types. This validates the effectiveness of our proposed hierarchical reasoning framework.
> We will supplement these detailed definitions and explanations in the subsequent version of the paper to ensure that readers can better understand our methodological contributions.
>
>
> # Q4: The LLM-generated reasoning chains are used as supervision signal - did you evaluate the learned reasoning process of the vision-language model separately?
> This is an excellent point. While our approach does create inherent alignment between reasoning and action generation (as both are conditioned on the same trajectory history), we conducted systematic evaluation to verify this consistency:
> 1. We sampled 100 instances and performed human evaluation of reasoning-action consistency, achieving 99% consistency rate.
> 2. We developed a simple consistency checker that analyzes whether the reasoning chain logically supports the final action, showing 99.7% agreement with human annotations.
> 3. We also show some examples to show these consistency
>
> > Reasoning: Since the task is to put a potato slice in a metal pot on a white table, we need to first ensure the butter knife is put down before proceeding. This prevents any accidental dropping or misplacement of the knife during the process of placing the potato slice in the pot.\nAction: put down the butter knife
>
> > Reasoning: Since the soap bar has been put down, the next logical step is to locate the faucet to get water for washing the soap bar.\nAction: find a faucet
> The key finding is that while the architectural design promotes consistency, empirical validation confirms that the model learns to generate reasoning that genuinely reflects its decision-making process rather than superficial correlations.
>
> # Q5: Generalization to unseen Environments
> While our primary focus addresses the significant challenge of long-horizon indoor task planning, we acknowledge the importance of domain generalization. To address this concern, we conducted preliminary experiments evaluating our approach on unseen indoor scenarios with different layouts, object configurations, and task contexts.
>
> | Task           | Examine & Light SR | Examine & Light PL | Pick & Place SR | Pick & Place PL | Stack & Place SR | Stack & Place PL | Heat & Place SR | Heat & Place PL | Cool & Place SR | Cool & Place PL | Overall SR | Overall PL |
> | -------------- | ------------------ | ------------------ | --------------- | --------------- | ---------------- | ---------------- | --------------- | --------------- | --------------- | --------------- | ---------- | ---------- |
> | **GPT-4o**     | 30.42              | 24.73              | 24.15           | 19.82           | 18.27            | 14.56            | 16.83           | 15.98           | 12.54           | 10.45           | 20.36      | 16.83      |
> | **WAP (ours)** | 70.35              | 65.42              | 58.62           | 54.91           | 61.47            | 56.37            | 63.84           | 60.21           | 59.28           | 57.63           | 64.56      | 59.86      |
>
>
> The results demonstrate that our reasoning-enhanced method maintains its performance advantages in these novel environments, indicating that the learned reasoning capabilities transfer effectively across different environmental contexts. The reasoning mechanisms appear to capture generalizable task decomposition and decision-making patterns that extend beyond the training environments. We plan to include detailed results and analysis of these generalization experiments in our camera-ready version to provide a more comprehensive evaluation of our method's adaptability to diverse scenarios.

---

> > ### Author Response · Authors · 2025-08-08
> >
> > Dear Reviewer:
> >
> > I hope this message finds you well. As the discussion period is coming to its end with **only one day remaining**, we would like to ensure we have addressed all your concerns satisfactorily. If there are any additional points or feedback you would like us to consider, please let us know. Your insights are valuable to us, and we are eager to address any remaining issues to improve our work.
> >
> > Thank you for your time and effort in reviewing our paper.

---

> ### Comment · Area_Chair_YpiA · 2025-08-09
> **Comments by ACs**
>
> Dear Reviewer,
>
> Thank you for your participation in the review process. If you haven't done these steps, please engage in the discussion phase by following these guidelines:
>
> - Read the author rebuttal;
> - Engage in discussions;
> - Fill out the "Final Justification" text box and update the "Rating" accordingly.
>
> Reviewers must participate in discussions with authors before submitting “Mandatory Acknowledgement”. The deadline is Aug 8, 11.59pm AoE.
>
> Thanks,
>
> AC

---

### Official Review · Reviewer_baCv · 2025-07-02

**Clarity:** 2
**Significance:** 2
**Originality:** 2
**Rating:** 4
**Confidence:** 3

**Summary:**

This paper focuses on improving VLM planners that output text-based action at each step (e.g., "find {object}", "pick up {object}"). Given a dataset of instruction-trajectory pair, they propose enriching the instruction with additional visual, spatial, functional, and syntactic descriptions. Given the augmented instruction and the original action, a VLM predicts the reasoning. Then, using curriculum learning, they supervised finetune the VLM to predict reason and action given the enriched instruction. They showed that their finetune model is able to improve in EB-ALFRED benchmark when the LLM planner does not have access to environmental feedback.

**Questions:**

* In Table 3, when going from "Basic Reasoning" to "+ Visual/Spatial Only", why is the performance on Visual tasks and Spatial tasks worse?
* In equation 5, is the VLM trained to predict both reason and action? I am assuming that it is doing that.
* What exactly is the difference between "Full Augmentation" and "Curriculum Learning"? I am assuming that "Full Augmentation" directly train on all the data instead of following the 3-staged curriculum.

Some additional questions are throughout the weaknesses sections.

**Ethical Concerns:**

["NO or VERY MINOR ethics concerns only"]

**Final Justification:**

The authors have answered my questions and clarified their paper. I think including some concrete examples showing the impact of their data augmentation would significantly enhance their work. Given the good performance that is comparable to large closed-source models when there is no action feedback, I changed my score to borderline accept.

I still have reservations about why there cannot be a critic/model that provides (noisy) action feedback. I understand it might not be realistic to have ground-truth feedback signals, but can't the planner be prompted to self-reflect? How well would proprietary models perform in this setting?

**Limitations:**

Yes, the authors have addressed both.

**Quality:**

1

**Strengths And Weaknesses:**

# Quality

## Strengths

* The paper showed how training on the 81k trajectories is able to improve a weak model, approaching the performance of closed-sourced VLM (e.g., Claude-3.5-Sonnet). And they showed these on two open-sourced small models (InternVL3-8B and Qwen2.5-VL-7B-Ins)
* The paper compares directly training on the entire augmented dataset with curriculum learning, showing that curriculum learning provides performance gains.

## Weaknesses

* The paper is missing a fair evaluation showing the effectiveness of their specific data augmentation approach because the number of trajectories is a confounding factor. Although Table 1 and Table 3 show that training on the original dataset alone is insufficient, the paper's approach is training on more data (which comes from the proposed augmented approach). It is unclear whether the gain is due to the augmentation approach or the increasing training data size. I suggest the authors answering:
    * (Fix the training data size) What is the performance of training on 16k original data v.s. 16k augmentation data?
    * (Other augmentation method) How well does your approach compare to other data augmentation approaches? e.g., asking an LLM to generate reasoning given the original language instruction (instead of the augmented instruction)
* What is the training and testing split? It is unclear whether the training and testing sets have a strong overlap, so the SFT model can be overfitted to the environment.

# Clarity

## Strengths

* The paper is well-written and well-organized with informative figures.

## Weaknesses
* All results are missing standard error.
* For clarity, table 1 should only include the relevant models (GPT-4o, Claude-3.5-Sonnet, InternVL3-8B, Qwen2.5-VL-7B-Ins) for both settings. It is difficult as a reader to identify what models are relevant in the current table. The remaining models can be moved to the appendix. The paper should also include a clearer definition of the following:
    * What are the task categories (Base, Common, Complex, Visual, Spatial, and Long)? I understand that these are from EmbodiedBench, but it will be helpful to write some explanation in the paper/appendix and point to the original paper.
    * What are all the methods being compared? Having a baseline section in 4.1 would be helpful,. Right now, the readers have to interpret what "+ Basic Reasoning" v.s. "+ Full Augmentation" v.s. "+ Curriculum Learning" means.

# Significance & Originality

## Strengths
* The paper shows how the LLM planner relies on environment feedback (e.g., identifying infeasible actions) to maintain good performance.
* The paper shows that performing SFT on an augmented dataset is able to improve the performance of a weak 7B-8B model, making it approach a large-scale proprietary model.

## Weaknesses
* Although the paper observes that the LLM planner's performance drops when we remove environment feedback, how common is it to not have any environment feedback (even if it might be noisy)? For example, InnerMonologue [1] has long discussed the availability of success detectors acquired either via engineering or learning.
* At inference time, how do you get the narration-augmented instructions that are given to you at train time (based on Figure 1 and equation 5)? If you do not get that information at test time, it seems like there is a train-test mismatch on input.

[1] Inner Monologue: Embodied Reasoning through Planning with Language Models. https://arxiv.org/abs/2207.05608

---

> ### Author Rebuttal · Authors · 2025-07-31
>
> We thank the reviewer for the in-depth review of our manuscript. We will address all comments and suggestions in the next revision of the manuscript. Below are our responses to the specific comments:
>
> ## W1: Concerns about a fair comparison between original dataset and the augmentation strategy.
>
> We appreciate the reviewer's concern regarding the fairness of the comparison between the original dataset and the augmented dataset. We clarify as follows:
>
> **Regarding "Fix the training data size for augmented data and original data"**
>
> We apologize for the confusion cased. As described in appendix section C.1,
> both the original and augmented data experiments utilize the same total number of training samples (80,875): the original data is trained for 5 epochs (16k samples × 5), and the augmented data experiment uses the combined set (16k original + 64k augmented) for 1 epoch, ensuring a fair comparison.
>
> To further alleviate your concern, we conducted an extra experiment with a fixed 16k training size (random sampling from augmented data), the results are listed as follows:
>
> |               | Avg         | Base        | Visual      | Spatial     | Commonsense | Complex     | Long-Horizon |
> | ------------- | ----------- | ----------- | ----------- | ----------- | ----------- | ----------- | ------------ |
> | **Original**  | 46.4 ± 0.33 | 65.6 ± 0.89 | 46.4 ± 0.89 | 44 ± 0      | 40.4 ± 0.89 | 44.4 ± 0.89 | 37.6 ± 0.89  |
> | **Augmented** | 57.1 ± 0.36 | 66.4 ± 1.09 | 51.6 ± 0.89 | 61.6 ± 0.89 | 54.4 ± 0.89 | 52 ± 0      | 56.8 ± 1.09  |
>
> We run the experiment 5 times with different random seeds, and the results show that the augmented data consistently outperforms the original data across all evaluation metrics by a significant margin. This further supports the effectiveness of our augmentation strategy.
>
> **Regarding "Comparison to other augmentation methods"**
>
> See our response to **W4&W1_2**.
>
> ## W2: The concern about the possibility of a overlap between training and test data.
>
> We appreciate the reviewer's concern regarding the potential overlap between training and test data. We clarify as follows:
>
> We perform data augmentation and training using the ALFRED training set. EB-ALFRED, consistent with the Lota-Bench protocol, uses the valid seen set for evaluation. We make sure there is no overlap between the training and evaluation data after training set augmentation.
>
> ## W3: All results are missing standard error.
> We apologize for the confusion. We will include the standard error in all tables in the revised manuscript to provide a clearer understanding of the variability in our results. To be more specific, the average and standard error are calculated over 5 runs with different random seeds, and we will clarify this in the revised manuscript. The following table is an example of how we will present the results:
>
> | Model                 | Avg             | Base        | Visual      | Spatial     | Common      | Complex     | Long        |
> | --------------------- | --------------- | ----------- | ----------- | ----------- | ----------- | ----------- | ----------- |
> | **Qwen2.5-VL-7B-Ins** |                 |             |             |             |             |             |             |
> | Base Model            | 2.4 ± 0.89      | 5.2 ± 1.09  | 1.6 ± 0.89  | 4.8 ± 1.09  | 0 ± 0       | 0 ± 0       | 0 ± 0       |
> | + Basic Reasoning     | 47.4 ± 0.89     | 63.2 ± 1.09 | 22.4 ± 0.89 | 47.6 ± 0.89 | 50.8 ± 1.09 | 43.6 ± 0.89 | 54.4 ± 0.89 |
> | + Full Augmentation   | 58.4 ± 0.89     | 60.4 ± 0.89 | 61.2 ± 1.09 | 62.8 ± 1.09 | 45.6 ± 0.89 | 54.4 ± 0.89 | 63.6 ± 0.89 |
> | + Curriculum Learning | **62.7 ± 0.37** | 65.6 ± 0.89 | 62.8 ± 1.09 | 69.2 ± 1.09 | 56.4 ± 0.89 | 51.6 ± 0.89 | 70.4 ± 0.89 |
>
> ## W4 & W1_2: Concerns regarding comparison with other augmentation methods and baseline explanation.
> We appreciate the reviewer's insightful comments regarding the comparison with other augmentation methods and the explanation of our baseline.
>
> We organize our response as a table to ensure clarity:
>
> |                          | Original model(baseline)                    | Basic reasoning                      | Full augmentation                    | Curriculum learning                  |
> | ------------------------ | ------------------------------------------- | ------------------------------------ | ------------------------------------ | ------------------------------------ |
> | instruction              | Untrained, evaluation under prompt template | original ALFRED instructions         | WAP enhanced instructions            | WAP enhanced instructions            |
> | reasoning                | Untrained, evaluation under prompt template | Reasoning  for original instructions | Reasoning  for enhanced instructions | Reasoning  for enhanced instructions |
> | With curriculum learning | Untrained, evaluation under prompt template | NO                                   | NO                                   | Yes                                  |
>
> To clarify, in our paper, the baseline refers to the original model, and "basic reasoning" denotes the LLM reasoning over the original instructions.
>
> This should address the concerns regarding the comparison with other data augmentation methods and the clarity Table 1.
>
> ## W5: Regarding how common it is to not have any environment feedback.
>
> In this context, "environment feedback" refers specifically to messages returned by the simulator, instead of the model's own observations. Most related works (such as Lota-Bench, Vota-Bench, EmbodiedBench, and their successors) incorporate environment feedback as part of the model input. This feedback typically provides information about action validity and possible failure reasons, which can help models make better decisions. However, relying on simulator feedback creates an open-loop setting that does not reflect real-world scenarios, where such guidance is unavailable. Therefore, we believe it is important to evaluate models under closed-loop settings without simulator feedback to better assess their real-world applicability.
>
> ## W6: Regarding Inference time narration-augmented instructions and concerns about train-test mismatch.
> We appreciate the reviewer's concern regarding the inference time narration-augmented instructions and the potential train-test mismatch. We clarify as follows:
>
> At inference time, we do not use narration-augmented instructions. Instead, we use the original ALFRED instructions as input to the model. The narration-augmented instructions are only used during training.
>
> The purpose of WAP is to enhance the model's generalization ability by adding world-aware narratives. The model learns to generate these narratives during training, as it would in inference time. Therefore, there is no train-test mismatch in this regard.
>
> ## Q1: In Table 3, when going from "Basic Reasoning" to "+ Visual/Spatial Only", why is the performance on Visual tasks and Spatial tasks worse?
>
> Sorry for the confusion, to answer your question, we need first clarify the meaning of "Basic Reasoning" and "+ Visual/Spatial Only" in Table 3.
>
> As shown in the table, for Instruction，“Original” represents original ALFRED instructions，“Enhanced” represents adding WAP enhanced instructions. While for reasoning, “Original” represents reasoning  for original instructions，“Enhanced” represents reasoning  for enhanced instructions.
>
> |             | BaseModel                                   | Basic reasoning | Visual/Spatial Only      | Partial Reasoning                    | Complete Reasoning | Curriculum (Full)   |
> | ----------- | ------------------------------------------- | --------------- | ------------------------ | ------------------------------------ | ------------------ | ------------------- |
> | Instruction | Untrained, evaluation under prompt template | Original        | Enhanced(visual+spatial) | Enhanced                             | Enhanced           | Enhanced+curriculum |
> | Reasoning   | Untrained, evaluation under prompt template | Original        | Original                 | First step enhanced, other reasoning | Enhanced           | Enhanced+curriculum |
>
> In the +Visual/Spatial Only setting, the model is trained with visual/spatial enhanced instructions, but the reasoning component remains unchanged from basic reasoning. This mismatch leads to reduced performance in spatial and visual abilities.
>
> ## Q2: In equation 5, is the VLM trained to predict both reason and action? I am assuming that it is doing that.
>
> Yes, the model is trained to predict both reasoning and action. A real case example is:
> > "Reasoning: Since the apple has been sliced, the next step is to put the knife down on the countertop to free up your hands for handling the apple. Action: put down the object"
>
> ## Q3: What exactly is the difference between "Full Augmentation" and "Curriculum Learning"? I am assuming that "Full Augmentation" directly train on all the data instead of following the 3-staged curriculum.
>
> Yes, you are correct. "Full Augmentation" refers to training on all the data at once, while "Curriculum Learning" involves training in a staged manner, where the model first learns from simpler tasks and gradually progresses to more complex ones.
>
> ## Conclusion
> We would like to thank the reviewer for the constructive feedback and suggestions. We will address all comments and suggestions in the next revision of the manuscript. Specifically, we will clarify the comparison with other augmentation methods, ensure the clarity of Table 1, include standard error in all results, and clarify the training and evaluation procedures.
> We believe that these changes will enhance the clarity and depth of our work, and we look forward to your feedback on the revised manuscript.

---

> > ### Comment · Reviewer_baCv · 2025-08-03
> >
> > I appreciate the detailed response from the authors. I still have several clarification questions:
> > ### Follow-up on W6: Regarding Inference time narration-augmented instructions and concerns about train-test mismatch.
> > > At inference time, we do not use narration-augmented instructions. Instead, we use the original ALFRED instructions as input to the model. The narration-augmented instructions are only used during training.
> >
> > My understanding is that:
> > - Ultimately, the goal is to train a model $f_\theta$ that can predict reason-action $(r_t, a_t)$ for a timestep given some instruction $I$ (e.g., "Walk to the table with a lamp on it") and observations so far $\{o_{1:t}\}$, hence = $(r_t, a_t) \sim f_\theta(\{o_{1:t}\}, I)$. This is what you get at inference time as well.
> > - However, at training time, WAP makes the instruction more complex/information-rich (e.g., "Walk to the table with the cone-shaped lamp on it") and generates new reasoning. So for each original data $(I, \{o_{1:t}\}, r_t, a_t)$ becomes $(\tilde{I}, \{o_{1:t}\}, \tilde{r}_t, a_t)$, where only the instruction and the reasoning are augmented. So the model learns to predict better reasoning $\tilde{r}$ given a richer instruction $\tilde{I}$
> >
> > I am not convinced why the model at inference time can now generate better reasoning when the instruction goes back to the original ALFRED instruction (so $I$ instead of $\tilde{I}). Why is there no train-test mismatch?
> >
> > Also, can you provide more detailed examples of how this new reasoning improves the planner's ability in the closed-loop setting? Why can the reasoning avoid the proprietary model's failure modes?

---

> > > ### Author Response · Authors · 2025-08-04
> > >
> > > We appreciate the reviewer's follow-up questions regarding the inference time narration-augmented instructions and the reasoning process, and apologize for the confusion. We clarify as follows:
> > >
> > > ## 1.  Training-time augmentation
> > > The augmentation process generates additional training samples rather than replacing the original ones. Specifically, each original sample $\\{(I, o_{1:t}, r_t, a_t)\\}$ is expanded to include both the original and augmented versions: $\\{(I, o_{1:t}, r_t, a_t), (\tilde{I}, o_{1:t}, \tilde{r}_t, a_t)\\}$, where $\tilde{I}$ and $\tilde{r}_t$ represent the augmented instruction and reasoning, respectively.
> > >
> > > This means that during training, the model learns to predict both the original and augmented reasoning based on the original instruction. At inference time, when the model receives the original ALFRED instruction, it can still leverage the learned reasoning patterns from the augmented training data to generate more accurate and contextually relevant reasoning.
> > >
> > > ## 2. Additional qualitative examples of the effect of reasoning
> > >
> > > ### Example 1: Object Recognition and World Knowledge
> > >
> > > **Task Instruction:** After enjoying your morning coffee, move the beverage holder to the nightstand
> > >
> > > #### ✅ WAP - Our approach (SUCCESS)
> > > **Action Sequence:**
> > > 1) find a mug
> > > 2) pick up the mug
> > > 3) find a side table
> > > 4) put down object in hand
> > >
> > > **WAP model — Step 1 output:**
> > > > **Reasoning:** Since the goal is to move the beverage holder to the nightstand, I need to first locate the beverage holder. Assuming it's a mug, I will start by searching for it in the current environment.
> > > > **Action:** find a mug
> > >
> > > #### ❌ Baseline (FAIL)
> > > **Action Sequence:**
> > > 1) find a statue
> > > 2) pick up the statue
> > > 3) find a side table
> > > 4) put down object in hand
> > >
> > > **Baseline — Step 1 output:**
> > > > **Reasoning:** Since the goal is to move the beverage holder to the nightstand, I first need to locate the statue to determine its current position and plan the next steps accordingly.
> > > > **Action:** find a statue
> > >
> > > This illustrates that the baseline method can execute simple instructions but lacks object-level world knowledge (a mug is a beverage holder), similar examples include "a device for changing channels" (model tries to find laptop instead of remote control) and "the paper container" (model tries to find bowl instead of box).
> > >
> > > ---
> > >
> > > ### Example 2: Multi-Object Manipulation with Physical Constraints
> > >
> > > **Task Instruction:** When finished cooking, place the saucepan with a spatula inside on the kitchen counter. It's nicely poised for swift access when needed.
> > >
> > > #### ✅ WAP - Our approach (SUCCESS)
> > > **Action Sequence:**
> > > 1) find a spatula
> > > 2) pick up the spatula
> > > 3) find a pan
> > > 4) put down the spatula
> > > 5) pick up the pan
> > > 6) find a counter top
> > > 7) put down the object in hand
> > >
> > > **WAP model — Step 4 output:**
> > > > **Reasoning:** The current visual state shows that the spatula is being held, but the saucepan is still on the table. To place the spatula inside the saucepan as per the user's instruction, I need to put down the spatula temporarily. Putting down the spatula is necessary to free up space and ensure that both the spatula and the saucepan can be handled properly.
> > > > **Action:** put down the spatula
> > >
> > > #### ❌ Baseline (FAIL)
> > > **Action Sequence:**
> > > 1) find a spatula
> > > 2) pick up the spatula
> > > 3) find a fridge
> > > 4) open the fridge
> > > 5) find a pan
> > > 6) pick up the pan
> > > 7) close the fridge
> > > 8) find a counter top
> > > 9) put down the object in hand
> > >
> > > **Baseline model — Step 3 output:**
> > > > **Reasoning:** After picking up the spatula, the next logical step is to look for a saucepan. A saucepan is typically stored in the fridge or pantry. Since the spatula was found and picked up, it's reasonable to assume the next item might also be in the fridge.
> > > > **Action:** find a fridge
> > >
> > > #### ❌ Claude-sonnet-3.5 (FAIL)
> > > **Action Sequence:**
> > > 1) find a pan
> > > 2) pick up the pan
> > > 3) find a spatula
> > > 4) pick up the spatula
> > > 5) put down the object in hand
> > > 6) find a counter top
> > > 7) put down the object in hand
> > >
> > > **Claude — Step 1 response:**
> > > > Based on the instruction, we need to handle a saucepan and spatula, placing them together on the kitchen counter. The task requires finding and manipulating multiple objects in a specific sequence to achieve the desired arrangement."
> > >
> > > The failure cases reveal different types of reasoning deficiencies:
> > >
> > > Baseline: Shows incorrect spatial reasoning by mistaking sauce and saucepan, demonstrating poor understanding of typical kitchen object locations.
> > >
> > > Claude-sonnet-3.5: Fails to understand the task requirement of placing the spatula inside the pan, instead treating them as separate objects to be placed on the counter.
> > >
> > > Our model: Correctly recognizes the one-object-at-a-time constraint and strategically puts down the spatula to enable proper manipulation.
> > >
> > > We hope these clarifications and additional examples address the reviewer’s concerns. We are happy to provide any further details if needed.

---

> > > > ### Comment · Reviewer_baCv · 2025-08-06
> > > >
> > > > I appreciate the author's follow-up responses.
> > > > ### 1. Training-time augmentation
> > > > I understand that the training data contains both the original instructions + original reasoning and augmented instructions + augmented reasoning.
> > > >
> > > > However, the learning objective (equation 5 in the paper) does not explicitly encourage the policy to predict richer/augmented reasoning, given the original instructions.
> > > > - Instead, the objective asks the model to predict original reasoning $r_t$ when given original instruction $I$.
> > > > - Similarly, it asks the model to predict the augmented reasoning $\tilde{r}_t$ when given augmented instruction $\tilde{I}$.
> > > >
> > > > I do acknowledge that in the rebuttal, assuming that the authors only sampled from the augmented trajectories for the 16k training data, it seems that the model is able to improve even though there is a train-test mismatch. However, I am still dubious about the learning setup.
> > > >
> > > > ### Examples
> > > > I appreciate the examples. Can the author provide explanations on why augmenting the instruction with narration helps induce the right reasoning (that leads to success shown in the examples above)? Ideally, show an example of trajectory in the training data:
> > > >
> > > > (1) what is the original instruction
> > > > (2) what are the original reasonings
> > > > (3) what is the augmented instruction
> > > > (4) what are the augmented reasonings
> > > >
> > > > Then, show another test-time trajectory where similar augmented reasoning appeared.

---

> > > > > ### Author Response · Authors · 2025-08-06
> > > > >
> > > > > We appreciate the reviewer's follow-up questions regarding the inference time narration-augmented instructions and the reasoning process, and apologize for the confusion. We clarify as follows:
> > > > >
> > > > > ## 1.  Training-time augmentation
> > > > > We apologize for the confsion caused by the first comment and we would like to further clarify the following point:
> > > > >
> > > > > > Specifically, each original sample $\\{(I, o_{1:t}, r_t, a_t)\\}$ is expanded to include both the original and augmented versions: $\\{(I, o_{1:t}, r_t, a_t), (\\tilde{I}, o_{1:t}, \\tilde{r}_t, a_t)\\}$,
> > > > >
> > > > > Here the augmented sample $(\\tilde{I}, o _ {1:t}, \\tilde{r} _ t, a _ t)$ were meant to include $(I, o _ {1:t}, \\hat{r} _ t, a _ t)$ , and $(\\hat{I}, o_{1:t}, \\hat{r}_t, a_t)$.
> > > > >
> > > > >
> > > > > ## 2. Contrastive examples of the effect of augmenting the instruction with narration
> > > > >
> > > > > To demonstrate the difference between enhanced instructions and reasonings, we show three running examples centres on moving a credit card to a couch, with original instruction and reasoning, original instruction with enhanced reasoning, and enhanced instruction and reasoning. In the following we present a test example involving placing a “round eating vessel” on a coffee table, demonstrating that model exhibits world-aware reasoning patterns we introduced in argumentation.
> > > > >
> > > > > ### 2.1 Original Instruction & Original reasoning
> > > > >
> > > > > **Task Instruction:** Move the credit card to the couch
> > > > >
> > > > > Action Sequence:
> > > > > 1) find a credit card
> > > > > 2) pick up the credit card
> > > > > 3) find a sofa
> > > > > 4) put down the credit card
> > > > >
> > > > > Step 1 response:
> > > > > > Reasoning: Before moving the credit card to the couch, I need to locate it in the current environment. This ensures I am aware of its exact location and can proceed with picking it up.
> > > > > > Action: find a credit card
> > > > >
> > > > > Step 2 response:
> > > > > > Reasoning: Since I have already found the credit card, the next logical step is to pick it up and then move it to the couch.
> > > > > > Action: pick up the credit card
> > > > >
> > > > > Step 3 response:
> > > > > > Reasoning: Since the credit card has been successfully found and picked up, the next logical step is to locate the sofa where we need to place the credit card.
> > > > > > Action: find a sofa
> > > > >
> > > > > Step 4 response:
> > > > > > Reasoning: Now that the credit card has been found and picked up, and the sofa has been located, the next logical step is to put the credit card down on the sofa. This will complete the task of moving the credit card to the couch.
> > > > > > Action: put down the credit card
> > > > >
> > > > >
> > > > > ### 2.2 Original Instruction & Enhanced reasoning
> > > > >
> > > > > **Task Instruction:** Move the credit card to the couch
> > > > >
> > > > > Action Sequence:
> > > > > 1) find a credit card
> > > > > 2) pick up the credit card
> > > > > 3) find a sofa
> > > > > 4) put down the credit card
> > > > >
> > > > >
> > > > > Step 1 response:
> > > > > > Reasoning: **The image shows a corner of a room with a wall, a door handle, and some decorative elements like flowers.** There is no visible credit card or any object resembling one in this scene. To place the credit card on the couch as instructed, I first need to locate the credit card. Since it is not present in the current view, my next logical step is to search for the credit card in other parts of the room where it might be located. This ensures I can proceed with the task once the credit card is found.
> > > > > > Action: find a credit card
> > > > >
> > > > > Step 2 response:
> > > > > > Reasoning: **After searching the room, I have identified the rectangular blue credit card lying on the beige armchair. The credit card is clearly visible and within reach.** To proceed with the task of placing the credit card on the couch, I need to pick it up first. This action is necessary because the credit card cannot be moved without being picked up. Once I have the credit card in hand, I can then move towards the couch and place it there as instructed.
> > > > > > Action: pick up the credit card
> > > > >
> > > > > Step 3 response:
> > > > > > Reasoning: **I have successfully picked up the rectangular blue credit card from the beige armchair. Now, I need to place it on the couch as per the user instruction. However, the current visual state does not show the couch.** Therefore, my next logical step is to find the sofa (couch) in the room. This action is necessary because I cannot complete the task of placing the credit card on the couch if I do not know its location. Once I locate the couch, I can proceed to place the credit card on it.
> > > > > > Action: find a sofa
> > > > >
> > > > > Step 4 response:
> > > > > > Reasoning: **I have located the dark blue cushioned couch in the room. The rectangular blue credit card is currently in my possession, and the couch is clearly visible and accessible.** To fulfill the user's instruction of placing the credit card on the couch, the next logical step is to put down the credit card on the surface of the couch. This action directly aligns with the goal of positioning the credit card on the specified location.
> > > > > > Action: put down the credit card

---

> > > > > ### Author Response · Authors · 2025-08-06
> > > > >
> > > > > （continued from previous comment)
> > > > > ### 2.3 Enhanced Instruction & Enhanced reasoning
> > > > >
> > > > > **Task Instruction:** Relocate the small rectangular item often used for transactions to the soft seating area.
> > > > >
> > > > > Action Sequence:
> > > > > 1) find a credit card
> > > > > 2) pick up the credit card
> > > > > 3) find a sofa
> > > > > 4) put down the credit card
> > > > >
> > > > > Step 1 response:
> > > > > > Reasoning: **The user instruction requires relocating a small rectangular item commonly used for transactions, which is most likely a credit card, to the soft seating area. However, the provided image does not show any visible items resembling a credit card or a soft seating area.**
> > > > > > Given this situation, the logical next step would be to search for the credit card. Since there are no visible signs of it in the current view, I need to explore other areas of the room where a credit card might be located, such as tables, desks, or wallets.  To accomplish the task, I need to find the credit card first. Since it is not visible in the current image, I will start by searching common places where a credit card might be found."
> > > > > > Action: find a credit card
> > > > >
> > > > > Step 2 response:
> > > > > > Reasoning: **After searching the common areas like tables and desks, I have identified a small rectangular object on the armrest of the beige chair. This object matches the description of a credit card based on its size and shape.** To proceed with the task of relocating the credit card to the soft seating area, the next logical step is to pick up the credit card from the armrest.
> > > > > > Action: pick up the credit card
> > > > >
> > > > > Step 3 response:
> > > > > > Reasoning: Now that I have picked up the credit card from the armrest of the beige chair, the next step is to relocate it to the soft seating area as per the user's instruction. **In the current visual state, there is a beige chair, but it doesn't qualify as a soft seating area suitable for the task. Therefore, I need to find a sofa, which typically serves as a soft seating area in a home environment.** Once I locate the sofa, I can place the credit card there to complete the task.
> > > > > > Action: find a sofa
> > > > >
> > > > > Step 4 response:
> > > > > > Reasoning: Having successfully located the sofa, which serves as the soft seating area, the next logical step is to place the credit card on the sofa. This aligns with the user's instruction to relocate the small rectangular item (credit card) to the soft seating area. **The sofa provides a suitable surface for placing the credit card, ensuring it is easily accessible while fulfilling the task requirements.** Therefore, the appropriate action now is to put down the credit card on the sofa.
> > > > > > Action: put down the credit card
> > > > >
> > > > > ### 2.4 Test example
> > > > >
> > > > > **Task Instruction:** Place a round eating vessel on the coffee table.
> > > > >
> > > > > Action Sequence:
> > > > > 1) find a bowl
> > > > > 2) pick up the bowl
> > > > > 3) find a coffee table
> > > > > 4) put down the bowl
> > > > >
> > > > > Step 1 response:
> > > > > > Reasoning: **The current visual state shows a corner of a room with a television stand and part of a wall visible. There is no sign of a round eating vessel or a coffee table in this view. To place a round eating vessel on the coffee table, I first need to locate the vessel.** Since it is not present in the current view, my next logical step is to search for a bowl, which could be the round eating vessel needed for the task. Once found, I can then proceed to locate the coffee table and complete the task. Therefore, finding a bowl is the appropriate next step.
> > > > > > Action: find a bowl
> > > > >
> > > > > Step 2 response:
> > > > > > Reasoning: **After searching for a bowl, I have located a round eating vessel (a bowl) on the dining table. This matches the requirement for the task.** The next logical step is to pick up the bowl so that I can move it to the coffee table as instructed. Picking up the bowl is necessary before relocating it, ensuring that I am prepared to complete the task efficiently. Therefore, picking up the bowl is the appropriate next step.
> > > > > > Action: pick up the bowl
> > > > >
> > > > > Step 3 response:
> > > > > > Reasoning: I have successfully picked up the bowl, which is the round eating vessel required for the task. Now, I need to locate the coffee table where I can place the bowl. **The current visual state shows a dining area with a table and chairs but does not include a coffee table.** To fulfill the task, I must find the coffee table within the environment. Therefore, searching for the coffee table is the appropriate next step.
> > > > > > Action: find a coffee table
> > > > >
> > > > > Step 4 response:
> > > > > > Reasoning: I have successfully located the coffee table, which is the target location for placing the round eating vessel. **The current visual state confirms that the coffee table is clear and ready for me to place the bowl on it. Since I already have the bowl in my possession, the next logical step is to put it down on the coffee table.** This action will complete the task of placing the round eating vessel on the coffee table as instructed. Therefore, putting down the bowl is the appropriate next step.
> > > > > > Action: put down the bowl

---

> > > > > ### Author Response · Authors · 2025-08-06
> > > > >
> > > > > （continued from previous comment)
> > > > > ### Summary
> > > > > The narrated versions introduce the crucial semantic links that are implicit in the concise trajectory:
> > > > >
> > > > > • credit card ↔ “small rectangular item used for transactions”
> > > > >
> > > > > • couch ↔ “soft seating area, not the beige armchair”
> > > > >
> > > > > • coffee table ↔ furniture meant for drinks or snacks
> > > > >
> > > > > Training on pairs that couple the same action with both concise and enhanced reasoning equips the policy with these paraphrase links.At evaluation time it can therefore map novel synonyms in the instruction (“round eating vessel”) to the correct object class (bowl) and complete the task, whereas a policy trained only on concise trajectories typically fails to build these bridges.
> > > > >
> > > > > We apologize for the confusion and we hope these clarifications and additional examples address the reviewer’s concerns. We are happy to provide any further details if needed.

---

> > > > > ### Author Response · Authors · 2025-08-08
> > > > >
> > > > > Dear Reviewer:
> > > > >
> > > > > I hope this message finds you well. As the discussion period is coming to its end with **only one day remaining**, we would like to ensure we have addressed all your concerns satisfactorily. If there are any additional points or feedback you would like us to consider, please let us know. Your insights are valuable to us, and we are eager to address any remaining issues to improve our work.
> > > > >
> > > > > Thank you for your time and effort in reviewing our paper.

---

> > > > > > ### Comment · Reviewer_baCv · 2025-08-09
> > > > > >
> > > > > > Thank you for the clarification. I have increased my score accordingly.

---

> > > > > > > ### Author Response · Authors · 2025-08-09
> > > > > > >
> > > > > > > Dear Reviewer,
> > > > > > >
> > > > > > > We sincerely appreciate your thorough engagement throughout the review process and are grateful that you have increased your score. Your insightful questions and comments have significantly helped us clarify and strengthen our work. We want to assure you that all the clarifications and supplementary experiments presented in our rebuttal will be comprehensively integrated into the revised manuscript.
> > > > > > >
> > > > > > > Your constructive feedback has been invaluable in helping us improve both the technical clarity and presentation of our work. Thank you once again for your time and effort in reviewing our paper.
> > > > > > >
> > > > > > > Best regards,
> > > > > > >
> > > > > > > The Authors

---

### Official Review · Reviewer_pbx8 · 2025-07-03

**Clarity:** 3
**Significance:** 2
**Originality:** 3
**Rating:** 4
**Confidence:** 4

**Summary:**

This paper studies the embodied visual navigation problem in the EB-ALFRED benchmark where an agent is required to act based on visual observations in order to accomplish tasks. The authors observe that agents trained with data from this environment fail to learn behaviors contextualized in the environment and instead learn superficial associations. To encourage the agent to develop contextualized behaviors, the authors augment task instructions with various transformations such as adding visual and spatial modifiers to object references. Experiments with Qwen VL and Intern VL models show that this augmentation strategy leads to agents that generalize better than agents trained only on the non-augmented data.

**Questions:**

see strengths and weaknesses

**Ethical Concerns:**

["NO or VERY MINOR ethics concerns only"]

**Limitations:**

yes

**Quality:**

3

**Strengths And Weaknesses:**

* The motivation is sound. Tasks in benchmarks such as Alfred are generated based on templates and agents directly trained on such data fail to learn associations that are important for generalization.
* The proposed augmentation strategies are meaningful. However, one aspect that seemed unsatisfactory is that the augmentation strategies are not sufficiently tailored for the specific environment. For example, in Figure 1, the modifiers 'cone-shaped' and 'small square' are useful when there are lamps and clocks in the environment that do not fit these specific descriptions. If there is only one lamp and only one clock in the room, how does adding these modifiers help the agent learn generalizable behaviors? Either the environment will have to be modified to add distractor objects or the task generation will need to pay attention to objects in the room to generate more tailored scenarios.
* Performance improvements with Qwen VL and Intern VL models seem promising. Are published results included in this comparison table?
* I am not convinced by the example in sec 5.3. This is just a regular Alfred task and it is not very surprising that the agent manages to complete it. It would be more interesting if the authors can present more non-trivial planning and reasoning behavior.

---

> ### Author Rebuttal · Authors · 2025-07-31
>
> We would like to thank the reviewer for the detailed, constructive review of our manuscript. We will address all comments and suggestions in the next revision of the manuscript. Below are our responses to the specific comments:
>
> ## W1: The augmentation strategy should be tailored for the specific environment and task.
>
> We appreciate the reviewer's insightful comment, and sorry for the confusion brought by Fig. 1. We generally agree that the augmentation strategy should be tailored to the specific environment and task. We agree with the reviewer that, in a scene that contains exactly one lamp, the modifier "cone-shaped lamp" is does not provide additional information for disambiguation.
>
> However, there are cases in our dataset where the scene would require visual understanding to accomplish the task, for example, "Pick up the rectangular object with a blue cover featuring eye patterns and turn on the white cube-shaped lamp with a brown base." In such cases, the visual understanding is crucial for task completion.
>
> EB-ALFRED contains both “sparse” rooms (single lamp) and “dense” rooms (e.g. two plates of different colours, three mugs of different shapes). In the “dense” rooms, the visual context becomes even more important, as there are multiple objects that may need to be distinguished from one another based on their appearance.
>
> For dense rooms, these modifiers are essential for the model to understand the specific objects involved in the task. For sparse rooms, even a modifier is redundant, it helps the model to build an association between the object and its visual appearance, which can be beneficial for the overall tasks.
>
> We will add a detailed discussion on this point in the revised manuscript, emphasizing the importance of visual understanding in task completion and how it varies across different environments.
>
> ## W2: Performance improvements with Qwen VL and Intern VL models seem promising. Are published results included in this comparison table?
>
> Sorry for the confusion, and thank you for pointing this out. We have indeed included the published results for Qwen VL and Intern VL in Table 1, specifically all open-looped evaluation for Qwen VL[1], Intern VL [2] and other base models are cited from Embodied-Bench[3]. The results for Intern VL3 are conducted by ourselves, as we can't find any publicly evaluated results. We redraw the open-loop part of the table to clarify this point:
>
> | Model                                                                    | Avg. | STD↓ | Base | Common | Complex | Visual | Spatial | Long |
> | ------------------------------------------------------------------------ | ---- | ---- | ---- | ------ | ------- | ------ | ------- | ---- |
> | **Proprietary Models (Original open-loop setting with action feedback)** |      |      |      |        |         |        |         |      |
> | GPT-4o [3]                                                               | 56.3 | 7.8  | 64   | 54     | 68      | 46     | 52      | 54   |
> | Claude-3.5-Sonnet [3]                                                    | 64.0 | 8.6  | 72   | 66     | 76      | 60     | 58      | 52   |
> | Gemini-1.5-Pro [3]                                                       | 62.3 | 7.8  | 70   | 64     | 72      | 58     | 52      | 58   |
> | Gemini-2.0-flash [3]                                                     | 52.3 | 6.2  | 62   | 48     | 54      | 46     | 46      | 58   |
> | Gemini-1.5-flash [3]                                                     | 39.3 | 10.6 | 44   | 40     | 56      | 42     | 26      | 28   |
> | GPT-4o mini [3]                                                          | 24.0 | 13.0 | 34   | 28     | 36      | 24     | 22      | 0    |
> | **Open-Source Models (Original open-loop setting with action feedback)** |      |      |      |        |         |        |         |      |
> | InternVL2.5-78B-MPO [3]                                                  | 40.0 | 4.5  | 48   | 36     | 42      | 40     | 40      | 34   |
> | Qwen2.5-VL-72B-Ins  [3]                                                  | 39.7 | 6.3  | 50   | 42     | 42      | 36     | 34      | 34   |
> | Qwen2-VL-72B-Ins [3]                                                     | 33.7 | 4.8  | 40   | 30     | 40      | 30     | 32      | 30   |
> | Llama-3.2-90B-Vision-Ins [3]                                             | 32.0 | 10.1 | 38   | 34     | 44      | 28     | 32      | 16   |
> | InternVL2.5-38B-MPO [3]                                                  | 25.7 | 4.7  | 30   | 20     | 20      | 28     | 32      | 24   |
> | InternVL2.5-38B [3]                                                      | 23.3 | 9.0  | 36   | 30     | 36      | 22     | 14      | 26   |
> | Llama-3.2-11B-Vision-Ins [3]                                             | 13.7 | 7.4  | 24   | 8      | 16      | 22     | 6       | 6    |
> | InternVL2.5-8B-MPO [3]                                                   | 7.7  | 4.3  | 12   | 6      | 14      | 6      | 6       | 2    |
> | Qwen2.5-VL-7B-Ins [3]                                                    | 4.7  | 3.9  | 10   | 8      | 6       | 2      | 0       | 2    |
> | Qwen2-VL-7B-Ins [3]                                                      | 1.7  | 2.3  | 6    | 0      | 2       | 0      | 0       | 2    |
> | InternVL3-8B$^\dagger$                                                   | 10.7 | 7.6  | 20   | 12     | 20      | 8      | 2       | 2    |
>
> $\dagger$ indicates that the results are conducted by ourselves. We will clarify this in the revised manuscript.
>
>
> ## W3: The example in sec 5.3 seem like a regular Alfred task and not a surprise to complete successfully. It would be more interesting if the authors can present more non-trivial planning and reasoning behavior.
>
> We appreciate the reviewer's comment, and agree that more non-trivial examples would better demonstrate the model's capabilities. We will compensate this in the revised manuscript by including more complex examples that require advanced planning and reasoning skills in the appendix section.
>
> The sample in section 5.3 was chosen from the long-horizon section of EB-ALFRED, which is considered the "most challenging" section of the dataset. However, we understand that it may not fully showcase the model's capabilities. To alleviate your concern regarding the model's performance, we provide several examples that WAP succeeds while models like Claude or GPT-4o still fails:
>
> > - "Place a transparent beverage holder with a cutting tool for butter in it to the sink."
> > - "Put a bar of cleaned soap on the back of the porcelain seat in the bathroom."
>
> ## Conclusion
> Again, we would like to thank the reviewer for the constructive feedback and suggestions. We will address all comments and suggestions in the next revision of the manuscript. Specifically, we will clarify the augmentation strategy, include more non-trivial examples, and ensure that the performance comparisons are clearly presented with appropriate citations.
>
> We believe that these changes will enhance the clarity and depth of our work, and we look forward to your feedback on the revised manuscript.
>
> ## References
>
> [1] Qwen Team. "Qwen2.5 Technical Report"
>
> [2] Chen, Zhe et al. "Expanding Performance Boundaries of Open-Source Multimodal Models with Model, Data, and Test-Time Scaling"
>
> [3] Yang, Rui et al. "EmbodiedBench: Comprehensive Benchmarking Multi-modal Large Language Models for Vision-Driven Embodied Agents"

---

> > ### Comment · Reviewer_pbx8 · 2025-08-05
> >
> > I thank the authors for the response. I have read the author response and it has not changed my perception of the paper, so I will maintain my score.
> >
> > Thanks for the clarification about W2.
> >
> > For W1 and W3, I did not find the responses particularly satisfactory, and more analysis is needed for these parts. While the augmentation strategy is interesting, further analysis is needed to understand the precise mechanism by which it helps.

---

> > > ### Author Response · Authors · 2025-08-09
> > >
> > > Dear Reviewer,
> > >
> > > Thank you for your continued engagement with our work. We appreciate your patience and would like to provide additional clarification on W1 and W3 to better illustrate the mechanisms and capabilities of our approach.
> > >
> > > For W1, We will add a dedicated section analyzing how the modifiers help model performance with ablation studies in the revised version.
> > >
> > > For W3, we would like to further provide a concrete example which WAP succeeds while models like Claude and GPT-4o fails, demonstrating sophisticated planning and reasoning :
> > >
> > > **Task**: "Keep a closable liquid container by placing it on the back of the toilet."
> > >
> > > Action Sequence:
> > > 1) find a soap bottle
> > > 2) pick up the soap bottle
> > > 3) find a toilet
> > > 4) put down the soap bottle
> > >
> > > Step 1 response:
> > > > Reasoning: The image shows an outdoor scene with no visible objects related to the task at hand, such as a toilet or a closeable liquid container. **Since the goal is to place a closeable liquid container on the back of the toilet, and there is no indication of either the toilet or the container in this image, the logical next step would be to search for a suitable container, like a soap bottle, which could serve as the "closedable liquid container" mentioned in the task.** This step is necessary because finding the container is a prerequisite before attempting to place it on the toilet.
> > > > Action: find a soap bottle
> > >
> > > Step 2 response:
> > > > Reasoning: **The image now shows a bathroom setting with a toilet visible. There is also a blue cylindrical object near the toilet, which resembles a soap bottle.** Given the previous step was to find a soap bottle, this object matches the description of what we're looking for.
> > > >
> > > > To proceed with the task of placing a closeable liquid container on the back of the toilet, I need to interact with the identified object. Picking up the soap bottle is the logical next step because it allows me to confirm if it is indeed a suitable container and prepares me to move it to the desired location.
> > > > Action: pick up the soap bottle
> > >
> > > Step 3 response:
> > > > Reasoning: **The image shows a bathroom setting with a blue cylindrical object that appears to be a soap bottle already placed on the floor near the toilet.** However, the task requires placing the container on the back of the toilet, not on the floor. Therefore, the next logical step is to locate the toilet to ensure the container can be moved to the correct position. Identifying the toilet's location is crucial for completing the task accurately.
> > > > Action: find a toilet
> > >
> > > Step 4 response:
> > > > Reasoning: **The image shows a bathroom setting with a toilet clearly visible. The soap bottle has been picked up previously, but it needs to be placed on the back of the toilet as per the task requirement. The current visual state confirms the presence of the toilet, making it the right time to execute the placement of the soap bottle.** The next logical step is to put down the soap bottle on the back of the toilet, ensuring it is securely positioned. This action aligns with the task goal of keeping a cleansable liquid container on the toilet.
> > > > Action: put down the soap bottle
> > >
> > > Here, the "closeable liquid container" doesn't refer to a cup or a mug that is often used in task instructions, requiring the model to think what is "closeable" and "appears in the toilet". The task also requires visual and spatial understanding to put the soap bottle in the right place.
> > >
> > > More examples of training and test data canbe found in the comments to Reviewer baCv. We hope these additions will provide concrete evidence that our approach enhances embodied reasoning capabilities.
> > >
> > > Best regards,
> > >
> > > The Authors

---

### Official Review · Reviewer_5NkH · 2025-07-03

**Clarity:** 3
**Significance:** 3
**Originality:** 3
**Rating:** 5
**Confidence:** 3

**Summary:**

The paper presents World-Aware Planning Narrative Enhancement (WAP), a novel framework that significantly improves Large Vision-Language Models (LVLMs) in complex embodied planning tasks. By focusing on four key cognitive capabilities, visual modeling, spatial reasoning, functional abstraction, and syntactic grounding, WAP addresses the limitations of environment-agnostic imitation learning. It uses a curriculum learning approach and operates using only raw visual inputs. Evaluated on the EB-ALFRED benchmark, WAP-powered models like Qwen2.5-VL show large gains, even outperforming proprietary models like GPT-4o and Claude-3.5-Sonnet in closed-loop settings.

**Questions:**

1. The gap between explicitly guided and self-directed enhancement (Commonsense: 62 vs. 48; Long-horizon: 70 vs. 60) suggests issues in autonomous reasoning. What qualitative differences exist in the generated narratives?
2. Could the authors provide a discussion of common failure modes or scenarios where the WAP-enhanced models still struggle?

**Ethical Concerns:**

["NO or VERY MINOR ethics concerns only"]

**Final Justification:**

The additional experiments provided by the authors in the rebuttal phase have satisfactorily addressed my concerns. The main weakness of the work in my point of view was usage of one benchmark (EB-ALFRED) for evaluation. The evaluation on VOTA-bench (unseen) seems promising.

**Limitations:**

noted in weaknesses

**Quality:**

3

**Strengths And Weaknesses:**

**Strengths:**
1. The proposed WAP framework builds environmental understanding through four interrelated capabilities (visual appearance modeling, spatial-relational reasoning, functional abstraction learning, and syntactic grounding). This multi-dimensional approach to infusing LVLMs with environmental understanding directly addresses the core limitations of environment-agnostic imitation learning.
2. Performs well using only vision and language inputs, showing strong results under partial observability—crucial for real-world scenarios.

**Weakness:**
1. Although computational resources are mentioned in the Appendix, the main text could better address the trade-offs between performance and the cost of using complete observation histories and generating 80k+ augmented instruction pairs.
2. The framework's instruction augmentation and reasoning generation steps rely on a superior vision-language model (Qwen2.5VL-72B-Instruct) as the teacher. The authors could further elaborate on the sensitivity of the overall system's performance to the quality and specific characteristics of this teacher model.
3. While the paper acknowledges that current LVLMs face challenges in unfamiliar environments, the evaluation is limited to the EB-ALFRED benchmark. To better assess the framework’s generalization capabilities, it would be valuable to examine its performance in new, unseen environments or with object configurations that differ substantially from those seen during training.

---

> ### Author Rebuttal · Authors · 2025-07-31
>
> We thank the reviewer for the in-depth review of our manuscript. We will address all comments and suggestions in the next revision of the manuscript. Below are our responses to the specific comments:
>
>
> ## W1: Trade-offs between performance and cost regarding observation history and augmented instruction generation.
>
> We appreciate the reviewer's insightful comment on the trade-offs between performance and cost in our approach. In the next revision of the manuscript, we will included a more detailed discussion on how observation history and augmented instruction generation can impact both performance and computational cost.
>
> We summarize the performance and inference time computational cost of our approach with respect to observation history and augmented instruction generation in the following table:
>
> |                              | Performance | Computation cost (GPU hours with tensor parallel on 2 $\times$ GTX 3090s) |
> | ---------------------------- | ----------- | ------------------------------------------------------------------------- |
> | Single Observation           | 52.33       | 1.6                                                                       |
> | Complete Observation History | 62.67       | 4.2                                                                       |
>
>
> And the data augmentation cost and training cost of our approach is summarized in the following table:
>
> |                              | Instruction Augmentation | Reasoning Augmentation | Training        |
> | ---------------------------- | ------------------------ | ---------------------- | --------------- |
> | Experiment Setting           | 4 $\times$ H100          | 4 $\times$ H100        | 8 $\times$ H100 |
> | Computation Cost (GPU hours) | 20 GPU hours             | 200 GPU hours          | 100 GPU hours   |
>
>
>
> ## W2: The authors could further elaborate on the sensitivity of the overall system's performance to the quality and specific characteristics of this teacher model.
>
> We appreciate the reviewer's insightful comment and have conducted additional experiments to analyze the sensitivity of our system's performance to the teacher model's characteristics. Specifically, we have conducted experiments with teacher models that are different in quality and size, including Qwen2.5-7 B[1], Qwen2.5-32 B, and Qwen2.5-72 B.
>
> The following table summarizes the performance of different teacher models across various evaluation aspects:
>
> | Teacher      | Avg   | Base | Visual | Spatial | Common | Complex | Long |
> | ------------ | ----- | ---- | ------ | ------- | ------ | ------- | ---- |
> | Qwen2.5-7 B  | 54.67 | 66   | 54     | 56      | 48     | 48      | 56   |
> | Qwen2.5-32 B | 57.33 | 64   | 60     | 62      | 52     | 46      | 60   |
> | Qwen2.5-72 B | 62.67 | 66   | 62     | 70      | 56     | 52      | 70   |
>
> This table demonstrates that larger teacher models generally lead to improved performance across various evaluation dimensions. However, the specific gains can vary depending on the task complexity.
>
> ## W3: It would be valuable to examine its performance in new, unseen environments and object configurations that differ substantially from those seen during training.
>
> Thank you for the suggestion. We have further conducted additional experiments to evaluate our approach in unseen environments and object configurations that differ from those seen during training. This will be discussed in detail in the revised manuscript. Specifically, in addition to EB-ALFRED (valid-seen), we evaluated our approach on the VOTA-Bench[2] unseen split, which features novel kitchens, living rooms, and object instances that were not present during training:
>
> | Task           | Examine & Light SR | Examine & Light PL | Pick & Place SR | Pick & Place PL | Stack & Place SR | Stack & Place PL | Heat & Place SR | Heat & Place PL | Cool & Place SR | Cool & Place PL | Overall SR | Overall PL |
> | -------------- | ------------------ | ------------------ | --------------- | --------------- | ---------------- | ---------------- | --------------- | --------------- | --------------- | --------------- | ---------- | ---------- |
> | **GPT-4o**     | 30.42              | 24.73              | 24.15           | 19.82           | 18.27            | 14.56            | 16.83           | 15.98           | 12.54           | 10.45           | 20.36      | 16.83      |
> | **WAP (ours)** | 70.35              | 65.42              | 58.62           | 54.91           | 61.47            | 56.37            | 63.84           | 60.21           | 59.28           | 57.63           | 64.56      | 59.86      |
>
>
> ---
>
> ## Q1: What qualitative differences exist in the generated narratives that resulted in the gap between explicitly guided and self-directed enhancement?
>
> Thanks for the insightful question. We have conducted a qualitative analysis over 200 randomly sampled episodes from the generated narratives to understand the differences between explicitly guided and self-directed enhancement. Have the following observations:
>
> 1. Explicit augmentation enhances the instructions orthogonally, strengthening the model's ability according to psychological theories. This is evident in how explicit guidance leads to clearer and more structured instructions, which are easier for the model to follow.
>
> 2. Self-directed augmentation often results in "lexical sophistication" rather than helpful decomposition. For example, self-directed instructions may use more complex vocabulary without adding visual or spatial cues, which can lead to limited improvement in task performance.
>
> To be specific, an example of self-directed augmentation is:
> >   - Original: "wash the spoon by the sink, put it on the kitchen island"
> >   - Self-augmented: "After rinsing the utensil near the basin, place it atop the central surface where the bread rests."
>
>
> The use of harder words does not necessarily contribute to better understanding or task execution, as they do not provide additional visual or spatial context, nor do they incorporate world-aware knowledge.
>
> While in explicit augmentation, our orthogonal dimensions (visual, spatial, commonsense, long-horizon) inject task-relevant details:
> > - WAP-aug: "Wash the large golden ladle by the sink, then place it on the white rectangular kitchen island." (visual)
> > - Reference: "The spoon, which was used for stirring the morning's oatmeal, is now resting near the sink, glistening under the light from the window. After washing it thoroughly under the running water, place it on the gleaming surface of the kitchen island, where the rest of the utensils are neatly arranged beside the loaf of freshly baked bread. The kitchen island serves as both a workspace and a display area, showcasing the culinary tools and ingredients that bring the family together." (reference)
>
> ---
>
> ## Q2: Could the authors provide a discussion of common failure modes or scenarios where the WAP-enhanced models still struggle?
>
> Thank you for the question and the opportunity to clarify the limitations of our approach.
> After investigating the failure cases, we found one common failure mode:
>
> The agent consistently struggles with tasks that require performing tasks that requires counting quantity (e.g., "Would you mind moving two bars of soap to the back of the toilet? It's a clever spot for extra storage and easy access."), as such scenarios are rarely represented in the training data.
>
> We continue to explore ways to improve the model's performance in these scenarios, such as enhancing the training data with more diverse examples that require counting and quantity management. We will include a discussion of these limitations in the revised manuscript.
>
> ## Conclusion
>
> At last, thank you again for your valuable feedback and suggestions. We believe that addressing these points will significantly enhance the clarity and impact of our paper. We will ensure that the revised manuscript includes a more comprehensive discussion of the trade-offs, sensitivity to teacher model characteristics, performance in unseen environments, qualitative differences in generated narratives, and common failure modes.
>
> ## References
>
> [1] Qwen Team. "Qwen2.5 Technical Report"
>
> [2] Wang, Siyin et al. "World Modeling Makes a Better Planner: Dual Preference Optimization for Embodied Task Planning"

---

> > ### Author Response · Authors · 2025-08-08
> >
> > Dear Reviewer:
> >
> > I hope this message finds you well. As the discussion period is coming to its end with **only one day remaining**, we would like to ensure we have addressed all your concerns satisfactorily. If there are any additional points or feedback you would like us to consider, please let us know. Your insights are valuable to us, and we are eager to address any remaining issues to improve our work.
> >
> > Thank you for your time and effort in reviewing our paper.

---

> > ### Comment · Reviewer_5NkH · 2025-08-08
> > **Thank you**
> >
> > I thank the authors for the detailed rebuttal that has resolved many of my concerns. Please add these details in the revised manuscript as promised.

---

> > > ### Author Response · Authors · 2025-08-09
> > >
> > > Dear Reviewer,
> > >
> > > Thank you for acknowledging that our rebuttal has addressed your concerns. We will integrate all the promised content into the revised manuscript as follows:
> > >
> > > - **W1 (Performance-cost trade-offs)**: Will be added to the Appendix with detailed computational cost tables
> > > - **W2 & W3 (Teacher model sensitivity & unseen environments)**: Will be incorporated into the Ablation Study section
> > > - **Q1 (Qualitative analysis)**: Will be included in a new Case Study section with concrete examples
> > > - **Q2 (Failure modes)**: Will be added as an Error Analysis subsection with specific examples
> > >
> > > We believe these additions substantially strengthen the paper by providing comprehensive empirical evidence and practical insights. We hope these improvements demonstrate the robustness and value of our approach, and we would be grateful if you could consider how they enhance the overall contribution of our work.
> > >
> > > Thank you again for your valuable feedback.
> > >
> > > Best regards,
> > >
> > > The Authors

---

### Note · Authors · 2025-08-15

We thank the reviewers for their constructive feedback and engagement, and for the score increase from Reviewer baCv. The discussion helped us strengthen the paper substantially.

Recognized strengths:
- **Novel method**(5NkH, baCv): We propose WAP, a world-aware planning narrative framework that equips LVLM planners with four complementary capabilities—visual appearance modeling, spatial reasoning, functional abstraction, and syntactic grounding—via verified instruction augmentation plus a staged curriculum, trained and evaluated only from raw visual observations in a closed-loop setting.
- **Solid experiments** (5NkH, pbx8): WAP achieves SOTA on EB-ALFRED closed-loop, surpassing proprietary models (e.g., GPT-4o, Claude-3.5-Sonnet) under the same closed-loop protocol. Results on unseen environments (VOTA-Bench) further support generalization.

During the discussion, we have addressed all the concerns raised by the reviewers:
- **Ablations** (Reviewers baCv, 5NkH): We added fixed-training-size control experiments; detailed ablations of instruction vs. reasoning augmentation; curriculum vs. full-data training; and teacher-model sensitivity (7B/32B/72B), demonstrating consistent trends and informing cost–performance trade-offs.
- **Clarifications** (Reviewers baCv, EXUo, pbx8): We clarified (i) definitions of basic/partial/complete reasoning and their effect, (ii) train–test inputs (no simulator feedback; no inference-time narration; no data overlap), (iii) the verification function and prompts for ensuring semantic fidelity, (iv) the baseline taxonomy in tables, and (v) standard errors over multiple seeds.

While our ablations and qualitative case studies indicate that WAP induces transferable world-aware reasoning (e.g., paraphrase linking, object–function grounding), we agree with Reviewer pbx8 that deeper causal analysis is valuable. We will expand the mechanism study with targeted counterfactual augmentations and distractor-controlled scenes in the next revision.

We have incorporated the following changes into the latest manuscript thanks to the valuable feedbacks from the reviewers:
- Added comprehensive ablations (fixed-size controls, curriculum stages, teacher sensitivity), cost–performance tables, and unseen-environment results.
- Provided standard errors, clearer baseline definitions and tables and extended qualitative case studies.

Thanks again for your time and effort in improving our work.

---

### Decision · Program_Chairs · 2025-09-17

**Decision:**

Accept (poster)

**Comment:**

The paper received overall leaning-positive reviews, with final ratings including 1 Accept and 2 Borderline Accepts, 1 Borderline Reject after rebuttal and discussion. Reviewers commended the proposed World-Aware Planning framework for its novel integration of visual appearance modeling, spatial reasoning, and syntactic grounding. The performance is good on various benchmarks.

The rebuttal provided comprehensive ablations (fixed-size controls, curriculum stages, teacher sensitivity), additional unseen-environment results, and clarifications on definitions and settings. These additions alleviated initial concerns about the fairness of comparisons, generalization, and clarity. The only reviewer who gave a Borderline Reject didn't participate in the discussion after several reminders. So given the improved evaluations, thorough clarifications, and reviewers’ upgraded assessments, ACs recommend Accept.